# Glycyrrhizic Acid Inhibits High-Mobility Group Box-1 and Homocysteine-Induced Vascular Dysfunction

**DOI:** 10.3390/nu15143186

**Published:** 2023-07-18

**Authors:** Laura Kate Gadanec, Ulf Andersson, Vasso Apostolopoulos, Anthony Zulli

**Affiliations:** 1Institute of Health and Sport, Victoria University, Melbourne, VIC 3030, Australia; laura.gadanec@live.vu.edu.au (L.K.G.); vasso.apostolopoulos@vu.edu.au (V.A.); 2Department of Women’s and Children’s Health, Karolinska Institute, 17177 Stockholm, Sweden; ulf.andersson@ki.se; 3Immunology Program, Australian Institute for Musculoskeletal Science, Melbourne, VIC 3021, Australia

**Keywords:** atherogenesis, cardiovascular diseases, glycyrrhizic acid, high-mobility group box-1, homocysteine, hyperhomocysteinemia

## Abstract

Hyperhomocysteinemia (HHcy) worsens cardiovascular outcomes by impairing vascular function and promoting chronic inflammation via release of danger-associated molecular patterns, such as high-mobility group box-1 (HMGB-1). Elevated levels of HMGB-1 have recently been reported in patients with HHcy. Therefore, targeting HMGB-1 may be a potential therapy to improve HHcy-induced cardiovascular pathologies. This study aimed to further elucidate HMGB-1′s role during acute HHcy and HHcy-induced atherogenesis and to determine if inhibiting HMGB-1 with glycyrrhizic acid (Glyz) improved vascular function. Male New Zealand White rabbits (*n* = 25) were placed on either a standard control chow (CD; *n* = 15) or atherogenic diet (AD; *n* = 10) for 4 weeks. Rabbit serum and Krebs taken from organ bath studies were collected to quantify HMGB-1 levels. Isometric tension analysis was performed on abdominal aorta (AA) rings from CD and AD rabbits. Rings were incubated with homocysteine (Hcy) [3 mM] for 60 min to induce acute HHcy or rhHMGB-1 [100 nM]. Vascular function was assessed by relaxation to cumulative doses of acetylcholine. Markers of vascular dysfunction and inflammation were quantified in the endothelium, media, and adventitia of AA rings. HMGB-1 was significantly upregulated in serum (*p* < 0.0001) and Krebs (*p* < 0.0001) after Hcy exposure or an AD. Incubation with Hcy (*p* < 0.0001) or rhHMGB-1 (*p* < 0.0001) and an AD (*p* < 0.0001) significantly reduced relaxation to acetylcholine, which was markedly improved by Glyz. HMGB-1 expression was elevated (*p* < 0.0001) after Hcy exposure and AD (*p* < 0.0001) and was normalized after Glyz treatment. Moreover, markers of vascular function, cell stress and inflammation were also reduced after Glyz. These results demonstrate that HMGB-1 has a central role during HHcy-induced vascular dysfunction and inhibiting it with Glyz could be a potential treatment option for cardiovascular diseases.

## 1. Introduction

Cardiovascular diseases (CVDs), recognized as pathologies affecting the heart and vasculature, remain the leading cause of global mortality and morbidity [1]. An independent risk factor for CVDs is an elevated plasma concentration of homocysteine (Hcy), clinically termed hyperhomocysteinemia (HHcy) [2]. Hcy is a sulfur-containing, non-proteinogenic amino acid generated as an intermediate by-product during methionine metabolism [2]. Early identification of a correlation between HHcy and atherogenesis was reported in children with congenital defects disrupting correct methionine metabolism [3]. These individuals presented with severe HHcy (≥100 µmol/L), advanced atherosclerotic lesion formation [3] and, if left untreated, had a 50% chance of experiencing a catastrophic atherothrombotic event (e.g., cerebrovascular accident, myocardial infarction, or venous thromboembolism) by the age of 30 [3]. Moreover, a 5 µmol/L deviation from normal Hcy levels is associated with a 20% increased risk of CVD development [4]. In the general population, HHcy has been linked to a diet high in methionine-rich foods (e.g., eggs, dairy products, meat and poultry), vitamin B deficiencies (e.g., pyridoxine, folic acid and cobalamin), prescription medications and pre-existing comorbidities [5].

Endothelial dysfunction has been demonstrated as the earliest vasculature abnormality associated with HHcy [6] and atherogenesis [7], preceding morphological changes and atherosclerotic lesion formation. HHcy and atherogenesis promote endothelial dysfunction by activating mechanisms of oxidative stress, including reducing bioavailability of the potent vasodilator nitric oxide (NO) [8], uncoupling endothelium nitric oxide synthase (eNOS) [9], decreasing eNOS phosphorylation and activity [10], increasing formation of damaging reactive oxygen species (e.g., nitrotyrosine and peroxynitrite) and disturbing antioxidant scavenging and buffering systems (Figure 1A,B) [11,12]. These alterations drive endothelial homeostasis imbalance and diminish endothelium-dependent vasodilation by reducing acetylcholine (Ach) sensitivity [13].

Aggressive and unresolving inflammation is both a progenitor and consequence of oxidative stress and exacerbates endothelium injury and dysfunction [14]. Damaged cells release endogenous alarmin signals, referred to as danger-associated molecular patterns (DAMPs), to promote protective mechanisms of sterile inflammation by engaging with pattern recognition receptors [15]. However, continuous secretion of DAMPs is detrimental and results in a vicious cycle of unresolving and exacerbated inflammation and cell stress [15], which is central to the initiation and progression of HHcy and atherogenesis. Recently, the DAMP high-mobility group box-1 (HMGB-1) has been suggested as a major contributor to HHcy [16] and atherogenesis [17] pathology, as an elevated plasma HMGB-1 level has been reported in patients with HHcy [16] and individuals with atherosclerotic disease (Figure 1B) [18,19].

HMGB-1 has been shown to directly impair endothelium-dependent vasodilation in response to cumulative doses of Ach in mouse aorta [18], which may be due to HMGB-1 promoting endothelial cell inflammation through direct engagement with pattern recognition receptors (e.g., toll-like receptor (TLR) 2/4, triggering receptor expressed on myeloid cells-1 (TREM-1), and receptor for advanced glycated end-products (RAGE)) [20] and decreasing eNOS protein expression and NO production (Figure 1B) [18].

There has yet to be a commercially available antagonist that specifically targets HMGB-1, and the majority of literature relies on neutralizing antibodies and botanicals for effective HMGB-1 inhibition. The prodrug glycyrrhizin or glycyrrhizic acid (Glyz) is a naturally occurring triterpene glycoside that is extracted from the roots and stolons of the *Glycyrrhiza glabra* or licorice plant and has been identified as an HMGB-1 inhibitor due to its ability to directly engage with HMGB-1 and inhibit its pathogenic function [21,22]. Results from experimental and clinical studies have reported cardioprotective effects of Glyz, including anti-inflammatory, antioxidant, anti-thrombotic and anti-atherosclerotic [23]; however, its role during Hcy-induced pathology remains poorly understood. 

The purpose of the present study was to further elucidate the pathogenic role that HMGB-1 has during acute and chronic HHcy-induced vascular dysfunction and to evaluate Glyz (HMGB-1 inhibitor) as a potential therapeutic option to restore normal vascular function. To address the aims, abdominal aortae (AA) from male New Zealand White rabbits fed either a 4-week standard control chow (CD) or atherogenic diet (AD) were collected for ex vivo isometric tension myography organ bath studies and semi-quantitative immunohistochemistry analysis to investigate the effect that acute and chronic HHcy, HMGB-1 and Glyz treatment had on Ach-mediated relaxation and protein expression of markers associated with vascular dysfunction, inflammation and cell stress (i.e., eNOS, nitrotyrosine and tumor necrosis factor alpha (TNFα)). An enzyme-linked immunosorbent assay (ELSA) was also performed on serum and Krebs–Henseleit solution (Krebs) to determine the effect that acute and chronic HHcy had on systemic and isolated artery HMGB-1 release.

## 2. Materials and Methods

### 2.1. Antibodies, Chemicals, and Kits

HMGB-1 ELISA kit (rabbit) (Cat#OKWB00408) was purchased from Aviva Systems Biology (San Diego, CA, USA); anti-eNOS/NOS Type III (Cat#610297) purified mouse monoclonal antibody and 3,3′-diaminobenzidine (DAB) substrate kit (Cat#550880) were purchased from Becton Dickinson Biosciences (VIC, Australia); unconjugated mouse monoclonal TNF-α antibody (Cat#NB600-1422) was purchased from Novus Biologicals (CO, USA); recombinant human HMGB-1 (Cat#RDS1690HMB050) was purchased from R&D Systems (Minneapolis, MN, USA); acetylcholine (Cat#A6625), DL-homocysteine (Cat#H4628), glycyrrhizic acid ammonium salt from glycyrrhiza root (licorice) (Cat#50531), phenylephrine (Cat#P1250000) and monoclonal anti-HMGB-1 (Cat#WH0003146M8) and anti-3-nitrotyrosine (Cat#N5538) antibodies produced in mouse were purchased from Sigma Aldrich (St. Louis, MO, USA); and ImmPRESS HRP goat anti-mouse IgG polymer detection kit (Cat#MP-7452-50) was purchased from Vector Laboratories (Burlingame, CA, USA).

### 2.2. Animal Model and Ethics Approval

Male New Zealand White rabbits (*n* = 25) at 8–10 weeks of age were purchased from Nanowie Small Animal Production Unit (Belbrae, VIC, Australia) or Flinders City University (Adelaide, SA, Australia) and were housed at the Victoria University Werribee Campus Animal Facilities. Upon arrival, animals were given a 7-day acclimatization period and were maintained at a constant temperature of 21 °C and relative humidity level between 40 and 70%. Animals were kept on a 12 h day/night circadian rhythm cycle, and food and water were supplied ad libitum. All experimental procedures were conducted in accordance with the National Health and Medical Research Council ‘Australian Code of Practice for the Care and Use of Animals for Scientific Purposes’ (8th edition, 2013; https://www.nhmrc.gov.au/about-us/publications/australian-code-care-and-use-animals-scientific-purposes (accessed on 1 May 2023)) and were approved by Victoria University Animal Ethics Committee (VUAEC #14/005 and #17/013).

### 2.3. HHcy-Induced Atherogenesis Diet

Once rabbits reached 10 weeks of age, they were housed individually, and a number table was used to randomly allocate animals into different dietary groups. Rabbits received either a 4-week CD (*n* = 15) or AD (*n* = 10), which was supplemented with cholesterol (0.5%) + methionine (1%) + peanut oil (5%) (Cat#SF00-218; Specialty Feeds, Perth, WA, AUS). A 4-week AD, rich in methionine (the precursor to Hcy), cholesterol and saturated fat, in leporine models has been established by our laboratory as a physiologically relevant diet to mimic a natural human diet by promoting endothelial dysfunction and sporadic lesion formation within 4 weeks [24,25] and atherosclerotic plaque development in 12 weeks [26,27,28,29]. For the purpose of this study, we are interested in the role that HMGB-1 has during endothelial dysfunction and if treatment with Glyz is beneficial.

### 2.4. Sedation, Anesthesia, and Humane Dispatch

Animals were first sedated using a subcutaneous injection of medetomidine (0.25 mg/kg) at the ‘scruff’ or base of the neck and were anesthetized using isoflurane (4%). An incision was made at the lower abdomen and the subcutaneous tissue and lower abdominal muscles were dissected to expose the inferior vena cava. Rabbits were humanely dispatched by exsanguinating via inferior vena cava perforation. At this time, venous blood was collected from the site of perforation for HMGB-1 serum analysis. Exsanguination was allowed for 3 min or until loss of color and dilation of pupils was observed and the diaphragm was then cut signifying death of the animal. A T-tube was introduced distal to the aortic arch to allow flushing with cold oxygenated Krebs (118 mM, NaCl; 4.7 mM KCl; 1.2 mM MgSO_4_·7H_2_O; 1.2 mM KH_2_PO_4_; 25 mM NaHCO_3_; 11.7 mM glucose; and 1.25 mM CaCl_2_) (pH: 7.4). The AA was retrieved from each animal, and under a light microscope were cleaned of connective and adipose tissue and dissected into 2 mm rings in preparation for isometric tension myography and immunohistochemistry.

### 2.5. Serum and Krebs HMGB-1 Quantification

To determine if a 4-week AD increases systemic release of HMGB-1, serum was collected from CD (*n* = 5) and AD (*n* = 6) rabbits. Venous blood (10 mL) was collected at the site of inferior vena cava perforation, centrifuged at 3500 RPM for 10 min at 4 °C and the separated serum was collected and stored at −80 °C and was diluted (1:2) before being used in the ELISA kit. Additionally, to determine if acute or chronic HHcy alters the release of HMGB-1 from arteries ex vivo, Krebs was collected from individual organ baths containing AA rings from CD (*n* = 4) and AD (*n* = 4) rabbits and CD AA rings incubated with Hcy (*n* = 4) to mimic severe and acute HHcy. To do this, AA rings were immediately placed into organ baths (OB8, Zultek Engineering, VIC, AUS) containing 5 mL of Krebs and were maintained at 37 °C and continuously bubbled with 95% carbogen (95% oxygen; 5% carbon dioxide) to mimic a physiologically relevant environment and were left to acclimatize for 30 min. At this time, 3 mM of Hcy was added to some rings and 300 µL was then removed from organ baths containing CD, AD and Hcy AA rings every 15 min over a 60 min time period and were immediately stored at −80 °C. A 3 mM dose of Hcy was chosen for this study as we [30,31] and others [32,33] have reported its ability to invoke pathological states mimicking severe acute HHcy in arterial organ bath and cell culture studies. For quantification in standards, blanks, Krebs, and serum samples were prepared and added to pre-coated 96-well plates and incubated at 37 °C for 90 min. The biotinylated HMGB-1 detector antibody was added to each well after plates were washed and incubated at 37 °C for 60 min. The HRP-streptavidin conjugate was then added to wells and incubated at 37 °C for 30 min. Plates were then washed 5 times and the TMB substrate was added. Plates were placed in the dark at 37 °C for 15 min or until vibrant blue color was observed. The stop solution was then added, and the plates were read using a microplate spectrophotometer (Bio-Rad Laboratories, CA, USA). The relative OD_450_ for each sample was read using the equation, relative OD_450_ = well OD_450_—mean blank well OD_450_. The mean absorbance was calculated between each group and a standard curve was used to interpolate values of samples.

### 2.6. Incubations and Isometric Tension Analysis

AA rings were immediately and sequentially transferred into adjacent organ baths (OB8 and OB16, Zultek Engineering, VIC, AUS) filled with 5 mL of Krebs and were maintained at 37 °C and continuously bubbled with 95% carbogen (95% oxygen; 5% carbon dioxide). AA rings were allowed to acclimatize for 30 min and were then mounted between two metal organ hooks attached to force displacement transducers that were connected to a data acquisition program (PowerLab 16, ADInstruments, NSW, AUS) to record vasoactive responses [24]. Rings were twice stretched to a resting tension of 2 g, and allowed to equilibrate for 30 min. AA rings were then (i) left to rest for 120 min serving as control rings (CD: *n* = 15; AD: *n* = 10); (ii) incubated with 100 nM human recombinant HMGB-1 (rhHMGB-1: *n* = 5) for 60 min; (iii) incubated with 3 mM Hcy (Hcy: *n* = 15) for 60 min to induce endothelial dysfunction observed in severe and acute HHcy; (iv) incubated with 1µM Glyz (Glyz: *n* = 10; AD + Glyz: *n* = 10); or (v) pretreated with Glyz 60 min before a 60 min incubation with Hcy (Glyz + Hcy: *n* = 10). Upon completion of incubations, rings were constricted with 3 × 10^−7 0^ M phenylephrine. Once a plateau was achieved, vascular function was assessed by a dose response to Ach (10^−8^ M–10^−5^ M, half Log units), and relaxation responses were converted to a percentage.

### 2.7. Immunohistochemistry Technique and Semi-Quantification Analysis

After isometric tension analysis studies, tissues were placed in 4% paraformaldehyde (pH 7.4) for 24 h, transferred into paraffin embedding cassettes and refrigerated at 4 °C in 1× phosphate-buffered saline solution (pH 7.4) for 24 h. Tissues were then processed concomitantly in paraffin (Thermo Scientific Spin Tissue Processor Microm STP 120). After tissue processing, vessels were vertically embedded into blocks of paraplast and cut into 5 µm sections using a manual microtome. Ribbon sections were then placed into a 45 °C water bath containing dissolved gelatin, allowed to expand to original size, mounted onto slides and dried in an oven at 37 °C for 5 days. Tissues were de-paraffined and rehydrated using a xylene and ethanol gradient, and in Tris buffer (pH 7.4), as previously established in this laboratory [26,27]. Non-specific antigen sites were blocked using 1% goat serum for 60 min. The antibodies were added to the tissue (eNOS Type III/nitrotyrosine: 1:100 and HMGB-1/TNFα: 1:100) and incubated for 24 h. Tissues where then incubated with secondary antibodies for 60 min and DAB was added to the tissues for 15 min to stain for the antigen of interest. Tissues were then counterstained with hematoxylin, dehydrated, mounted and prepared for quantification analysis.

To determine the expression of proteins in the endothelium, media and adventitia, 6–8 photographs per ring were taken at ×400 and ×1000 magnification under an Olympus BX50 microscope. Each image was loaded into the Micro Computer Imaging Device (Interfocus, UK) analysis program (Appendix A), and the ribbon tool was selected to trace the endothelium (Appendix A) and the outline tool was used to trace the media (Appendix A) and adventitia (Appendix A). A specific range of color intensity and hue was selected to detect the expression of the DAB stain. The color intensity and proportional area were recorded and averaged for images of each individual ring. These values were recorded in a spreadsheet and the proportional intensity was calculated using the following equation, proportional intensity = (1/color intensity) × proportional area. The proportional intensity was then averaged for each tissue. Due to loss of tissue during paraffin processing, block cutting and slide mounting, an *n* = 5–10 for immunohistochemistry studies was retrievable.

### 2.8. Statistical Analysis

GraphPad prism version 9.5.1 was used to analyze data from isometric tension, immunohistochemistry studies and ELISA kit assays. The significant *p* value was set at *p* < 0.05. An unpaired *t*-test (serum analysis), one-way ANOVA (immunohistochemistry quantification) or two-way ANOVA (isometric tension analysis and Krebs analysis) followed by Sidak’s multiple comparisons post hoc test was performed to determine significance. All data are represented as mean ± standard error of the mean (SEM).

## 3. Results

### 3.1. Serum HMGB-1 Concentration Is Significantly Elevated in Rabbits Fed a 4-Week AD

Rabbits fed a 4-week AD had significantly higher serum HMGB-1 concentration when compared to rabbits fed a CD (CD: 22.06 ± 0.80 ng/mL vs. AD: 44.19 ± 0.68 ng/mL, *p* < 0.0001) (Figure 2A), suggesting that a diet high in methionine, cholesterol and saturated fat may contribute to vascular dysfunction and atherogenesis by increasing systemic release of HMGB-1.

### 3.2. HMGB-1 Is Released by AA Rings in Response to Hcy Exposure

Krebs taken from organ baths containing AA rings that were incubated with Hcy had significantly elevated HMGB-1 concentrations 15 min (*p* < 0.01) to 60 min (*p* < 0.0001) after Hcy incubation when compared to Krebs from organ baths containing CD AA rings with no incubation (Figure 2B, Table 1). Similarly, Krebs collected from organ baths containing AA rings from rabbits fed an AD exhibited markedly increased HMGB-1 levels at 15 min (*p* < 0.05) and 60 min time points (*p* < 0.5) when compared to CD rings (Figure 2B, Table 1). HMGB-1 concentrations were also significantly greater 45 min (*p* < 0.001) and 60 min (*p* < 0.0001) after Hcy incubation when compared to AD (Figure 2B, Table 1). These results demonstrate that Hcy incubation directly triggers blood vessels to liberate HMGB-1 into the extracellular environment. Additionally, a 4-week AD stimulates blood vessels to secrete, which may contribute to increased systemic levels of HMGB-1.

### 3.3. Incubation with rhHMGB-1 or Hcy Cause Similar Reductions in Ach-Mediated Relaxation

Incubation with Hcy markedly reduced Ach-mediated relaxation at all doses; Ach 10^−8^ M (*p* < 0.05) to 10^−5^ M (*p* < 0.0001) when CD AA rings were compared (Figure 3A, Table 2). Similar results were observed after incubation with rhHMGB-1, as relaxation was significantly diminished in response to Ach from 10^−7.5^ M (*p* < 0.001) to 10^−5^ M (*p* < 0.0001) (Figure 3A, Table 2). Remarkably, nearly identical reductions in relaxation at each Ach dose were noted between Hcy and rhHMGB-1, as no significance was observed between relaxation responses at any Ach doses when comparing AA rings incubated with Hcy or rhHMGB-1 (Figure 3A, Table 2). These results show the almost identical detrimental effect that a 60 min incubation with either Hcy or rhHMGB-1 has on the ability of AA to relax to cumulative doses of Ach by diminishing Ach sensitivity and thus suggest that HMGB-1 may play a major role during acute Hcy exposure.

### 3.4. Inhibiting HMGB-1 with Glyz Enhances Ach-Induced Relaxation after Acute or Chronic HHcy

Pretreatment with Glyz was able to block the detrimental effect of Hcy and markedly improved relaxation to Ach from 10^−7^ M (*p* < 0.001) to 10^−5^ M (*p* < 0.0001) (Figure 3B, Table 3). Unexpectedly, Glyz treatment alone resulted in reduced relaxation to Ach at doses 10^−7 5^ M (*p* < 0.05), 10^−6 5^ M (*p* < 0.05), 10^−5 5^ M (*p* < 0.05) and 10^−5 0^ M (*p* < 0.05) (Figure 3B, Table 3). Rabbits fed a 4-week AD, supplemented with high methionine, cholesterol and fat to promote HHcy-induced endothelial and vascular dysfunction and initiate atherogenesis, displayed significantly impaired relaxation abilities from Ach 10^−8^ M (*p* < 0.0001) to 10^−5 0^ M (*p* < 0.0001), when compared to CD AA rings (Figure 3C, Table 4). Strikingly, treatment with Glyz was able to significantly enhance relaxation responses to Ach in AD AA rings from 10^−7 5^ M (*p* < 0.001) to 10^−5^ M (*p* < 0.0001) (Figure 3C, Table 4). Taken together, these results demonstrate the ability of Glyz, a known HMGB-1 inhibitor, to reduce the detrimental vascular effect of Hcy and enhance Ach sensitivity; however, Glyz alone impaired Ach-mediated vasodilation in CD AA rings, thus suggesting that inhibiting HMGB-1 is beneficial to restore vascular function during acute HHcy but may be disadvantageous in healthy blood vessels. 

### 3.5. Glyz Blocks Increased Expression of HMGB-1 after Hcy Exposure or a 4-Week AD

Low basal levels of HMGB-1 were detectable in the different arterial layers of CD AA rings; however, incubation with Hcy resulted in markedly elevated peptide expression in the endothelium (CD: 6.79 × 10^−3^ ± 1.90 × 10^−3^ PI vs. Hcy: 0.84 ± 0.12 PI, *p* < 0.0001) (Figure 4A,B,G), media (CD: 1.57 × 10^−4^ ± 3.65 × 10^−5^ PI vs. Hcy: 0.21 ± 0.03 PI, *p* < 0.001) (Figure 4A,B,H) and adventitia (CD: 5.38 × 10^−3^ ± 3.34 × 10^−3^ PI vs. Hcy: 1.37 ± 0.14 PI, *p* < 0.0001) (Figure 4A,B,I). Strikingly, pre-treatment with Glyz prevented increased expression of HMGB-1 induced by Hcy in the endothelium (Hcy vs. Glyz + Hcy: 0.01 ± 2.28 × 10^−3^ PI, *p* < 0.0001) (Figure 4B,D,G), media (Hcy vs. Glyz + Hcy: 4.00 × 10^−4^ ± 1.19 × 10^−4^ PI, *p* < 0.0001) (Figure 4B,D,H) and adventitia (Hcy vs. Glyz + Hcy: 1.35 × 10^−3^ ± 3.68 × 10^−4^ PI, *p* < 0.0001) (Figure 4B,D,I).

Rabbits fed a 4-week AD had significantly elevated HMGB-1 protein levels in the endothelium (CD vs. AD: 0.25 ± 0.04 PI, *p* < 0.0001) (Figure 4A,E,G) and media (CD vs. AD: 0.43 ± 0.07 PI, *p* < 0.0001) (Figure 4A,E,H). No statistical difference in HMGB-1 expression was observed in the adventitia between AD and CD AA rings (Figure 4A,E,I). AD AA rings treated with Glyz showed significantly reduced peptide expression of HMGB-1 in the endothelium (AD vs. AD + Glyz: 8.97 × 10^−3^ ± 2.39 × 10^−3^ PI, *p* < 0.05) (Figure 4E,F,G) and media (AD vs. AD + Glyz: 4.01 × 10^−3^ ± 1.12 × 10^−3^, *p* < 0.0001) (Figure 4F,H). While Glyz was able to reduce HMGB-1 expression in the adventitia, no statistical difference was noted (Figure 4E,F,I).

Interestingly, acute exposure to Hcy resulted in increased expression of HMGB-1 in the endothelium (AD vs. Hcy, *p* < 0.0001) (Figure 4B,E,G) and media (AD vs. Hcy, *p* < 0.0001) (Figure 4B,E,H). Whereas chronic elevated levels of Hcy, induced by a 4-week AD, resulted in higher peptide concentration HMGB-1 in the media (Hcy vs. AD, *p* < 0.0001) (Figure 4B,E,I).

### 3.6. Inhibiting HMGB-1 Increases eNOS Expression after Hcy Incubation

High expression of eNOS was observed in all AA layers collected from CD rabbits (Figure 5A,H–J); however, 60 min exposure to Hcy resulted in drastic reduction in the endothelium (CD: 0.49 ± 0.05 PI vs. Hcy: 9.58 × 10^−3^ ± 1.66 × 10^−3^ PI, *p* < 0.0001) (Figure 4B,H), media (CD: 0.44 ± 0.04 PI vs. Hcy: 3.97 × 10^−3^ ± 1.26 × 10^−3^ PI, *p* < 0.001) (Figure 5B,I) and adventitia (CD: 1.06 ± 0.09 PI vs. Hcy: 6.14 × 10^−3^ ± 2.14 × 10^−3^ PI, *p* < 0.001) (Figure 5B,J). Similarly, a 60 min incubation with rhHMGB-1 also resulted in potent reduction in the endothelium (CD vs. rHMGB-1: 0.12 ± 0.06 PI, *p* < 0.0001) (Figure 5C,I), media (CD vs. rHMGB-1: 2.92 × 10^−3^ ± 0.01, *p* < 0.01) (Figure 5C,H) and adventitia (CD vs. rhHMHB-1: 0.03 ± 0.01 PI, *p* < 0.0001) (Figure 5C,J), and while increased protein expression was observed in the media, no statistical relevance was shown. No differences were observed between Hcy and rHMGB-1 treated rings.

Pre-treatment with Glyz was able to enhance eNOS expression in the endothelium (Hcy vs. Glyz + Hcy: 0.22 ± 0.04 PI, *p* < 0.05) (Figure 5E,H) and the adventitia (Hcy vs. Glyz + Hcy: 0.34 ± 0.07 PI, *p* < 0.0001) (Figure 5E,J), but not in the media (Figure 5E,I).

eNOS expression within all layers of blood vessels between CD and AD was not statistically different (Figure 5F,H–J). Moreover, no difference in eNOS expression was observed in the endothelium and media of AD and AD + Glyz treated rings (Figure 5G–I); however, AD + Glyz treated rings had reduced eNOS expression in the adventitia (AD: 0.92 ± 0.07 vs. AD + Glyz: 0.18 ± 0.06, *p* < 0.0001) (Figure 5F,J).

Acute exposure to Hcy had significantly reduced eNOS expression in the endothelium (Hcy vs. AD: 0.36 ± 0.06, *p* < 0.0001) (Figure 5B,F,H), media (Hcy vs. AD: 0.46 ± 0.13 PI, *p* < 0.001) (Figure 5B,F,I) and adventitia (Hcy vs. AD, *p* < 0.0001) (Figure 5B,F,J) when compared to rabbits fed a 4-week AD to initiate the development of HHcy-induced atherogenesis.

### 3.7. Inhibiting HMGB-1 with Glyz Reduces Nitrotyrosine Expression after Hcy Exposure or a 4-Week AD

Minimal nitrotyrosine expression was noted in all AA layers collected from CD rabbits (Figure 6A,H–J); however, 60 min exposure to Hcy resulted in significant upregulation of nitrotyrosine observed in the endothelium (CD: 3.70 × 10^−3^ ± 6.59 × 10^−3^ PI vs. Hcy: 0.32 ± 0.04 PI, *p* < 0.0001) (Figure 6B,H), media (CD: 9.78 × 10^−4^ ± 3.27 × 10^−4^ PI vs. Hcy: 0.19 ± 0.03 PI, *p* < 0.0001) (Figure 6B,I) and adventitia (CD: 6.39 × 10^−3^ ± 2.38 × 10^−3^ PI vs. Hcy: 0.38 ± 0.05 PI, *p* < 0.05) (Figure 6B,J). Similarly, a 60 min incubation with rhHMGB-1 also resulted in significant augmented nitrotyrosine expression in the endothelium (CD vs. 0.19 ± 0.04 PI, *p* < 0.01) (Figure 6C,I) and adventitia (CD vs. 0.61 ± 0.13 PI, *p* < 0.01) (Figure 6C,J), and while increased protein expression was observed in the media, no statistical relevance was shown. Interestingly, exposure to rhHMGB-1 resulted in a greater increase of nitrotyrosine in the adventitia than Hcy (*p* < 0.01).

Strikingly, pre-treatment with Glyz to inhibit HMGB-1 was able to block upregulated expression of nitrotyrosine in the endothelium (Hcy vs. Glyz + Hcy: 0.01 ± 4.00 × 10^−3^ PI, *p* < 0.0001) (Figure 6E,H) and media (Hcy vs. Glyz + Hcy: 0.02 ± 8.15 × 10^−3^ PI, *p* < 0.0001) (Figure 6E,I), but showed no statistical significance in the adventitia even though reduced expression was reported.

A 4-week AD to induce chronic elevated levels of Hcy resulted in marked nitrotyrosine expression in the endothelium (CD vs. 0.46 ± 0.03 PI, *p* < 0.0001) (Figure 6E,H), media (CD vs. 0.29 ± 0.03 PI, *p* < 0.0001) (Figure 6E,I) and adventitia (CD vs. 0.99 ± 0.13 PI, *p* < 0.0001) (Figure 6E,J) when compared to CD AA rings. AD rings treated with Glyz showed drastically reduced expression of nitrotyrosine in the endothelium (AD vs. AD + Glyz: 0.08 ± 0.03 PI, *p* < 0.0001) (Figure 6F,H) and media (AD vs. AD + Glyz: 0.02 ± 0.01 PI, *p* < 0.0001) (Figure 6F,I) but had no effect on adventitial expression.

Expectantly, when comparing the effect that an acute and chronic HHcy model had on nitrotyrosine expression, we found that nitrotyrosine expression was significantly higher after a 4-week AD in the endothelium (Hcy vs. AD, *p* < 0.01) (Figure 6B,F,H) (Hcy vs. AD, *p* < 0.0001), media (Hcy vs. AD, *p* < 0.01) (Figure 6B,F,I) and adventitia (Hcy vs. AD, *p* < 0.001) (Figure 6B,F,J) when compared to AA rings incubated with Hcy for 60 min.

### 3.8. Inhibiting HMGB-1 with Glyz Reduces TNFα Expression after Hcy Exposure or a 4-Week AD

Exposure to Hcy resulted in increased expression of TNFα in the endothelium (CD: 1.74 × 10^−3^ ± 3.53 × 10^−4^ PI vs. Hcy: 0.17 ± 0.02 PI, *p* < 0.0001) (Figure 7A,B,H) and the adventitia (CD: 1.52 × 10^−3^ ± 6.56 × 10^−4^ PI vs. Hcy: 0.34 ± 0.06 PI, *p* < 0.0001) (Figure 7A,B,J). While Hcy did induce increased TNFα levels in the media, no significant differences were noted when compared to CD rings (Figure 7A,B,H). Interestingly, while rhHMGB-1 was able to cause moderate increases in TNFα, no statistical relevance was observed in any of the arterial layers (Figure 7C,H–J).

Pre-treatment with Glyz to inhibit HMGB-1 prior to Hcy exposure was able to inhibit increased TNFα expression in the endothelium (Hcy vs. Glyz + Hcy: 4.50 × 10^−3^ ± 5.54 × 10^−4^ PI, *p* < 0.001) (Figure 7E,H) and adventitia (Hcy vs. Glyz + Hcy: 1.46 × 10^−3^ ± 7.97 × 10^−3^ PI, *p* < 0.0001) (Figure 7E,J). Again, while there was a decline of TNFα protein in the media, no significance was observed with Glyz treatment (Figure 7E,I).

TNFα expression was markedly increased in the endothelium (CD vs. AD: 0.33 ± 0.06 PI, *p* < 0.0001) (Figure 6A,F,H), media (CD vs. AD: 0.39 ± 0.08 PI, *p* < 0.0001) (Figure 7A,F,I) and adventitia (CD vs. AD: 0.21 ± 3.58 × 10^−3^ PI, *p* < 0.0001) (Figure 7A,F,J) of AA rings from AD rabbits when compared to CD rabbits. Additionally, treatment of AD rings with Glyz was able to radically reduce TNFα expression in the endothelium (AD vs. AD + Glyz: 7.53 × 10^−3^ ± 3.17 × 10^−3^ PI, *p* < 0.0001) (Figure 7F,G,I), media (AD vs. AD + Glyz: 6.49 × 10^−3^ ± 2.37 × 10^−3^ PI, *p* < 0.0001) (Figure 7F,G,H) and adventitia (AD vs. AD + Glyz: 1.33 × 10^−3^ ± 4.76 × 10^−4^ PI, *p* < 0.0001) (Figure 7F,G,J).

Finally, we report that chronic HHcy caused by a 4-week AD significantly increases expression of TNFα in the endothelium (Hcy vs. AD, *p* < 0.001) (Figure 7B,F,H) and media (Hcy vs. AD, *p* < 0.0001) (Figure 7B,F,I) when compared to AA rings incubated with Hcy. However, the opposite effect is observed in the adventitia, as a 60 min incubation with Hcy significantly augments peptide expression of TNFα when compared to AD AA rings (Hcy vs. AD, *p* < 0.05) (Figure 7B,F,J).

## 4. Discussion

This study was designed to further elucidate the pathogenic role of HMGB-1 during acute HHcy and HHcy-induced atherogenesis. Measurement of plasma HMGB-1 determined that rabbits fed a 4-week AD, supplemented with methionine and cholesterol to promote HHcy-induced atherogenesis, had significantly elevated levels of HMGB-1. Furthermore, quantification of HMGB-1 in Krebs samples demonstrated increased release of HMGB-1 in response to Hcy incubation and a 4-week AD. Isometric tension myography revealed that HMGB-1 decreased sensitivity to Ach and vasoactive responses were consistent to those observed after Hcy incubation. Strikingly, treatment with Glyz to inhibit HMGB-1 activity was able to markedly enhance relaxation in models of acute HHcy or after a 4-week AD, reduced expression of nitrotyrosine and TNFα, upregulated eNOS levels and restored levels of HMGB-1 in blood vessels to those of rabbits fed a CD.

HMGB-1 has gained notoriety as a potential clinical biomarker to predict disease activity and severity, as elevated serum or plasma levels have been associated with multiple pathologies, including autoimmune diseases (e.g., systemic lupus erythematosus [34] and type I diabetes [35]), metabolic disorders (e.g., type II diabetes [36]) and CVDs (e.g., coronary [19,37] and peripheral artery [38] disease). Studies evaluating atherosclerotic disease progression in non-diabetic patients with coronary artery disease [19] and diabetic patients with coronary artery stenosis [37] or peripheral artery disease [38] reported markedly elevated serum HMGB-1 levels. Similar results have been replicated in animal models investigating the pro-atherogenic role of HMGB-1. Apolipoprotein E-deficient mice fed a high lipid diet [39,40,41] and microminipigs [42] fed a diet enriched with lipids and cholesterol displayed significantly greater serum levels of HMGB-1 when compared to wild-type controls or animals fed a standard chow diet. Importantly, neuralization of HMGB-1 in apolipoprotein E knockout mice reported attention of atherosclerotic disease, shown by reduced atherosclerotic lesion size and decreased accumulation of pro-inflammatory cells and mediators [39]. Moreover, a positive correlation between HHcy and HMGB-1 has recently been suggested, as patients with moderate HHcy and 8-week-old male Sprague Dawley rats fed a 4-week high methionine diet (2%) have increased plasma HMGB-1 concentrations [16]. Results from this study were consistent with published data suggesting a direct relationship between HHcy and HMGB-1 [16], as our 4-week AD markedly increased systemic release of HMGB-1 as shown by elevated plasma HMGB-1 concentration.

The proposed mechanisms driving augmented systemic release of HMGB-1 involve (a) mass sustained self-release by damaged vascular (e.g., endothelial, medial and adventitial) and recruited immune cells (e.g., macrophages and monocytes), and (b) passive release by necrotic endothelial cells via pyroptosis (a newly defined mechanism of lytic programmed cell death, triggered by pro-inflammatory mechanisms) [38,43]. Under an acute high Hcy condition (150 µM for 24 h), shuttling of HMGB-1 from the nucleus into the cytoplasm for release into the extracellular environment has been observed in cultured human umbilical vein endothelial cells (HUVECs) [16], and is initiated by endothelial cell injury, inflammation and pyroptosis [20]. However, release and upregulation of HMGB-1 is not limited to the endothelium, as increased protein expression has been reported within the medial and adventitial layers during established atherosclerotic disease [44,45,46]. Rat-derived smooth muscle cells from thoracic aorta have been shown to respond to the chemoattractant ability of HMGB-1, resulting in vascular smooth muscle proliferation and migration, alterations in cell shape, cytoskeleton reorganization and restenosis of blood vessels [47]. A positive correlation between increased HMGB-1 serum levels and enhanced protein expression in endothelial and vascular smooth muscle cells, atherosclerotic plaques and inflammatory cells located in the media and adventitia has been demonstrated in patients with abdominal aortic aneurysm [44]. Additionally, Hcy exposure induces inflammation, proliferation, migration and phenotypic changes in vascular smooth muscle cells [45], and exacerbates adventitia inflammation by activating adventitial fibroblasts [46]. In this study, Krebs samples obtained from organ baths revealed increased secretion of HMGB-1 from isolated AA rings subsequent to a 60 min incubation with Hcy or a 4-week AD. Furthermore, immunohistological assessment revealed elevated expression of HMGB-1 within the endothelium, media and adventitia after acute or chronic HHcy and was reversed in rings treated with Glyz. These results suggest that Hcy directly upregulates HMGB-1 expression in the vasculature and increases release of HMGB-1 into the extracellular environment, resulting in augmented systemic levels.

Glyz is a naturally occurring bioactive triterpene glycoside conjugate of glucuronic acid and glycyrrhetinic acid, and is recognized by the Generally Recognized As Safe branch of the Food and Drug Administration [48]. Due to its safety profile, Glyz’s botanical application is extensive, including as a natural sweetener in confectionary goods and an emulsifier and gel-forming ingredient in cosmetics [48]. Importantly, Glyz has been widely investigated for its diverse anti-inflammatory, antihypertensive, antioxidative, antimicrobial and antithrombotic pharmacological properties, which may be attributed to its ability to engage with HMGB-1 [48]. Evidence from in silico studies have demonstrated the ability of Glyz to sequester and directly interact with HMGB-1 to inhibit its pathogenic function and engagement with innate immune receptors, such as TLR2/4, TREM-1 and RAGE [48,49]. Results from nuclear magnetic resonance and fluorescence studies have reported the ability of Glyz to directly associate with two shallow concave surfaces on HMGB-1 in a dose-dependent manner, resulting in the hindrance of its cytokinetic, chemoattractant and mitogen activating abilities [49,50]. Furthermore, Glyz’s profound ability to suppress HMGB-1 release from injured cells may also be explained by its ability to inhibit HMGB-1 phosphorylation [51] and expression [52]. Incredibly, results obtained from our immunohistochemistry studies demonstrate the ability of Glyz to abolish increased expression of HMGB-1 within the endothelium, media and adventitia in AA rings either exposed to Hcy or from rabbits fed a 4-week AD. Importantly, this finding suggests that Glyz may be able to enter cells and interfere with production and degradation steps that are central to HMGB-1 biology and release. This hypothesis is supported by a study involving cerebral vasospasm secondary to subarachnoid hemorrhage in rats, which reported a reduction of intracellular peptide and mRNA expression of HMGB-1 in rat basilar artery after Glyz treatment [53]. Similarly, a complementing study has shown that in murine macrophages (RAW 264.7) challenged with lipopolysaccharide (a TLR4 activating lipoglycan present on the outer membrane of Gram-negative bacteria), treatment with Glyz was able to prevent shuttling of HMGB-1 from the nucleus into the cytoplasm and blocked upregulation of HMGB-1 mRNA [52].

Endothelial cells are vulnerable to the damaging effect of Hcy [16] and HMGB-1 [18], which can promote endothelial dysfunction and initiate atherogenesis [18,26,27,28]. Reduced sensitivity to Ach and decreased capacity to release endothelium-derived dilating factors, predominantly NO produced by eNOS, are hallmarks of endothelial dysfunction observed in HHcy [54] and atherogenic pathology [7,55]. Interestingly, we report no differences between eNOS expression in CD and AD rings. While this finding appears to contradict previous works that report reduced eNOS in atherogenesis, studies from our laboratory have shown this phenomenon and suggest that eNOS overexpression in atherogenic rabbits is a compensatory mechanism and may not correlate with increased activity, phosphorylation or synthesis of NO [24,26,27,28]. HMGB-1 has also been shown to directly impair endothelium-dependent vasodilation in response to Ach in wild-type mouse aorta, but does not affect vascular smooth muscle cell-dependent dilation to sodium nitroprusside [18]. Results from our isometric tension studies show an almost identical reduction in vascular relaxation after incubation with Hcy or rhHMGB-1. Furthermore, when HMGB-1 was inhibited by Glyz, relaxation was improved in rings from AD rabbits or after acute Hcy exposure. The ability of Glyz to enhance endothelium-dependent relaxation to Ach during vascular dysfunction has recently been noted in the literature, as a study evaluating the protective vascular effect of Glyz reported significantly greater relaxation responses of aorta rings to Ach dose-response in diabetic rats treated with Glyz [56]. Reduced eNOS expression may explain decreased Ach sensitivity and subsequent impaired vasodilation, as eNOS peptide expression was significantly lower in rings exposed to Hcy or rhHMGB-1. Interestingly, we show that treatment with Glyz was able to enhance eNOS expression in the endothelium and adventitia of AA rings after Hcy incubation but not in AA rings from AD rabbits. These results align with a study that noted the anti-atherosclerotic properties of Glyz in male New Zealand White rabbits fed a 4-week high cholesterol (1%) and fat (6%) diet before being reverted to a 4-week low cholesterol (0.025%) diet to induce atherosclerosis, as rabbits with Glyz-eluting stents fitted on the left carotid artery demonstrated increased eNOS expression, promoted reendothelialization, enhanced luminal diameter and inhibited intimal hyperplasia [57]. Future studies utilizing Glyz should investigate the direct effect that it has on vasoactive responses of blood vessels by performing a dose-response and determine its ability to alter eNOS phosphorylation and activity. Taken together, these results support the hypothesis that HMGB-1 has a fundamental role during Hcy-induced vascular pathologies by diminishing vasoactive responses, which may be in part due to a reduction in eNOS protein expression or eNOS activity and phosphorylation, resulting in reduced NO release. However, while these results support a detrimental function of HMGB-1, recent literature has highlighted HMGB-1 as an important regulator of endothelium-specific activities (i.e., angiogenesis, migration, endothelium-dependent relaxation, and production of free radicals), and when genetically absent from the endothelium, reports of disadvantageous systemic effects have been documented [58]. Mice with endothelium-specific HMGB-1 deletion have increased mean arterial blood pressure, augmented plasma HMGB-1 concentration and decreased Ach-mediated endothelium dilation, which were partially improved by HMGB-1 overexpression [58]. Additionally, reductions in NO, eNOS phosphorylation (serine^1177^) and eNOS protein expression were also observed in HMGB-1 endothelium-specific knockout mouse aortae and HUVECs transfected with HMGB-1 silencing RNA [58], thus suggesting that intracellular and extracellular HMGB-1 have opposing effects on endothelial cells, and targeting extracellular HMGB-1 may improve endothelial dysfunction during HHcy and atherogenesis by reducing inflammation and cellular stress.

Nitrotyrosine is a stable end-product biomarker of oxidative and nitrosative stress, generated when peroxynitrite (a highly reactive and damaging free radical formed by NO and superoxide) incorporates with protein-bound or free tyrosine residues [59]. Nitrotyrosine has consistently been regarded as a major driver of endothelial and vascular dysfunction in HHcy and atherogenic pathology by reducing NO bioavailability and eNOS activity, promoting oxidative and cellular stress, activating pro-inflammatory pathways, and accelerating atherosclerotic lesion formation and plaque rupture [59,60,61]. While there appears to be a relationship between peroxynitrite, nitrotyrosine and HMGB-1, the interlinking mechanism(s) remains poorly understood. A positive correlation between elevated plasma nitrotyrosine and HMGB-1 levels in acute ischemic stroke patients, increased co-localized protein expression of HMGB-1 and nitrotyrosine in ischemic rat brain tissue and augmented HMGB-1 levels in cultured rat brain microvascular endothelial cells incubated with 3-morpholino-syndonimine (a peroxynitrite donor; 500 µM) [62] have recently been described. Similar results were demonstrated in ischemic rat myocardial tissue, as upregulated tissue expression of HMGB-1 and nitrotyrosine were noted [63]. However, this study also reported augmented HMGB-1 concentrations in medium from cultured cardiomyoblasts and primary cardiac myocytes and non-myocytes (e.g., endothelial, vascular smooth muscle cells, fibroblasts and cardiac progenitor cells) from neonatal mice incubated with peroxynitrite (100 µM or 250 µM, 4 h) [63]. Intriguingly, the authors show that the increase of HMGB-1 in cell medium was associated with a profound loss of cellular HMGB-1 by necrotic cells, but cells undergoing apoptosis retained their cellular HMGB-1 expression profile [63]. Therefore, suggesting that necrotic and stressed cells passively release HMGB-1, resulting in elevated extracellular HMGB-1 levels due to intracellular liberation and not from upregulated gene or protein expression [63]. Taken together, peroxynitrite may be an important mediator of HMGB-1 activation, expression and release, and increased nitrotyrosine expression after rhHMGB-1 incubation may directly result in nitrosative and oxidative stress; however, more research is needed to further establish the interlinking mechanisms.

Results from our immunohistochemistry experiments demonstrate that inhibiting HMGB-1 with Glyz was able to drastically reduce the expression of nitrotyrosine in all layers of AA rings incubated with Hcy and in the endothelium and media of rabbits fed a 4-week AD to promote HHcy-induced atherogenesis. This may be attributed to Glyz reducing oxidative and cell stress and promoting antioxidative mechanisms that attenuate nitrotyrosine formation. In male CD1 mice that underwent spinal cord surgery to produce injury at the thoracic level via extradural compression, pre-treatment (30 min before spinal cord injury) and post-treatment (1 h and 6 h after spinal cord injury) with Glyz (10 mg/kg) resulted in significantly reduced nitrotyrosine expression in the spinal cord 24 h after injury [64]. Similar results have also been reported in mice with hepatotoxicity induced by acetaminophen, as treatment with Glyz (80 mg/kg) 90 min after acetaminophen administration was able to provide hepatoxicity protection by decreasing hepatic mitochondrial and hepatocellular damages (indicated by serum analysis of glutamate dehydrogenase, alanine aminotransferase and aspartate aminotransferase activity), attenuated mitochondrial swelling and distortion, reducing necrosis of hepatic cells, inhibiting neuronal nitric oxide synthase activity and mRNA expression and decreased liver protein tyrosine nitration and nitrotyrosine levels [65]. Furthermore, Glyz treatment (10 mg/kg) 5 min prior to carrageenan pleural cavity injection to induce pleurisy was able to attenuate lipid peroxidation and immune cell infiltration and decreased upregulation of intracellular adhesion molecules (e.g., P-selectin and intracellular adhesion molecule 1) and nitrotyrosine in lung tissue [66]. While the ability of Glyz to attenuate Hcy-induced pathologies has yet to be validated, results from our study are supported by others that have determined Glyz’s ability to alter nitrotyrosine expression making it a promising therapeutic option for CVDs.

TNFα is an immunomodulating pleiotropic pro-inflammatory cytokine that is released upon DAMP (e.g., HMGB-1) and pattern-recognition receptor (e.g., TLRs) interaction to promote mechanisms of sterile inflammation in an effort to neutralize pathogenic infiltration [67]. However, its continuous upregulation and release has been positively correlated with Hcy-induced pathologies and atherogenesis [68,69,70]. Moreover, TNFα stimulates the release and binding of HMGB-1 to TLR4 in human aortic endothelial cells [67], which results in a vicious cycle of DAMP release, pattern recognition activation and cytokine release, promoting aggressive and exacerbated inflammation. Our results show that Hcy incubation or an AD resulted in upregulation of HMGB-1 and TNFα and may contribute to endothelial and vascular dysfunction by promoting inflammatory pathways [68,69,70]. However, while exposure to rhHMGB-1 did increase TNFα peptide expression in AA layers, there was no statistical significance. This may be explained by the release of TNFα requiring a longer incubation time with rhHMGB-1, as a study using human microvascular endothelial cells observed an increase in TNFα 3 h incubation with HMGB-1 [71].

The anti-inflammatory effect of Glyz is well established, and multiple studies highlight using models in which its ability to suppress the expression immune receptors (e.g., TLR2/4 and RAGE) and their downstream signaling molecules is shown [23]. Results obtained from immunohistochemistry studies show the ability of Glyz to potently reduce expression of TNFα in the different layers of AA rings incubated with Hcy or from rabbits fed a 4-week AD. Similar results have been reproduced in diabetic rats with atherosclerosis (induced by high fat diet and low dose streptozotocin), as a 16-week treatment of Glyz was able to significantly reduce serum and aorta levels of TNFα and other pro-inflammatory cytokines (i.e., interleukin-1β and -6) as well as decreased atherosclerotic plaque lesion size and intima thickness [72]. Conducive results have also been demonstrated in rats with coronary microembolization-induced injury, as pre-treatment with Glyz ameliorated myocardial dysfunction by preventing myocardial necrosis, downregulating mRNA expression of HMGB-1, inducible nitric oxide synthase, tumor necrosis factor alpha, interleukin -1β and -6 and suppressed stimulation of the HMGB-1/TLR4/nuclear factor-kappa beta pathway [73]. While the relationship between HMGB-1, Glyz and Hcy remains poorly understood, there is evidence that Glyz could be beneficial during Hcy-induced pathologies [74]. A study utilizing HUVECs (CRL-1730) noted that Hcy directly causes endothelial cell injury by effecting lectin-like oxidized-low-density lipoprotein receptor-1 DNA methylation through stimulation of TLR4/nuclear factor-kappa beta/DNA methyltransferase [74]. While the authors did not investigate production of pro-inflammatory cytokines, this study provides a rationale that TLR4 may have a prominent role in Hcy-induced endothelial cell dysfunction by activating TLR4 and resulting in the release of TNFα. 

Collectively, our results show a protective role for Glyz via neutralizing/inhibiting pathogenic extracellular HMGB-1, giving merit to its potential health benefits during Hcy-induced pathologies. However, while we and others have shown the cardiovascular protective abilities of Glyz, its use may be limited, as life-threatening cardiovascular (e.g., hypertension), neurological and renal (e.g., hypervolemia) side effects, electrolyte imbalances (e.g., hypokalemia, hypernatremia, pseudohyperaldosteronism), allergic reactions, and drug interactions have been reported in case-studies subsequent to excessive licorice ingestion [75,76]. Subsequent to oral ingestion, Glyz (the main constituent of licorice) is converted to 3β-monoglucuronyl-18β-glycyrrhetinic acid (3MGA) and the aglycone 18β-glycyrrhetinic acid (GA) by specialized intestinal bacteria β-glucuronidases [77]. Excessive licorice intake has been shown to induce physiological effects similar to the hormones aldosterone and cortisol, as 3MGA and GA bear molecular resemblance to intrinsic steroid-like structures, allowing direct interaction with mineralocorticoid receptors (MR) on the distal tubules of nephrons [78]. Activation of MR results in increased sodium and water resorption and potassium excretion; however, due to the low affinity that 3MGA and GA have for the MR compared to aldosterone, this theory has been debated [78]. Alternatively, it is more widely accepted that 3MGA and GA alter cortisol metabolism by inhibiting the type II isozyme of 11β-hydroxysteroid dehydrogenase (11β-HSD2) [79], a selective expressed (e.g., endothelial and vascular smooth muscle cells, cardiac cells and on the distal tubules of nephrons) enzyme responsible for converting inert/inactive keto-products to active cortisol and vice versa to prevent MR overactivation [75,76]. Inhibition of 11β-HSD2 by Glyz and its metabolites results in mineralocorticoid excess in renal tissue and increased vascular tone, exasperated contraction to vasopressor hormones and eNOS reduction in the vascular wall [80], and may be the reason as to why reduced vasodilation to Ach was noted in CD AA rings treated with Glyz. Moreover, prolonged 11β-HSD2 by excess licorice consumption has been correlated with hypertension, hypernatremia, hypokalemia, hypervolemia and hypoaldosteronemia, which can lead to serious life-threatening complications in patients with pre-existing CVDs [75,76]. Therefore, as there has yet to be a synthetic compound that can inhibit extracellular HMGB-1 without being hazardous to other organ systems, investigating the receptors, such as TLRs, RAGE or TREM-1 and their downstream molecules that HMGB-1 interacts with may reveal novel therapeutic targets for CVDs.

## 5. Conclusions

Herein, we report that the DAMP, HMGB-1, has a central role during acute HHcy and HHcy-induced atherogenesis by reducing Ach-mediated dilation, downregulating eNOS and upregulating nitrotyrosine and TNF-α. Inhibiting HMGB-1 with Glyz was able to improve Ach-mediated relaxation, abolished HMGB-1 expression and reduced levels of nitrotyrosine and TNF-α and increased eNOS during acute or chronic HHcy (Figure 8). However, due to the potential adverse effects of Glyz when consumed in excess, clinical studies are required to determine a safe daily intake dosage, and new pharmaceutical targets, such as receptors that interact with HMGB1, should be investigated.

## Figures and Tables

**Figure 1 nutrients-15-03186-f001:**
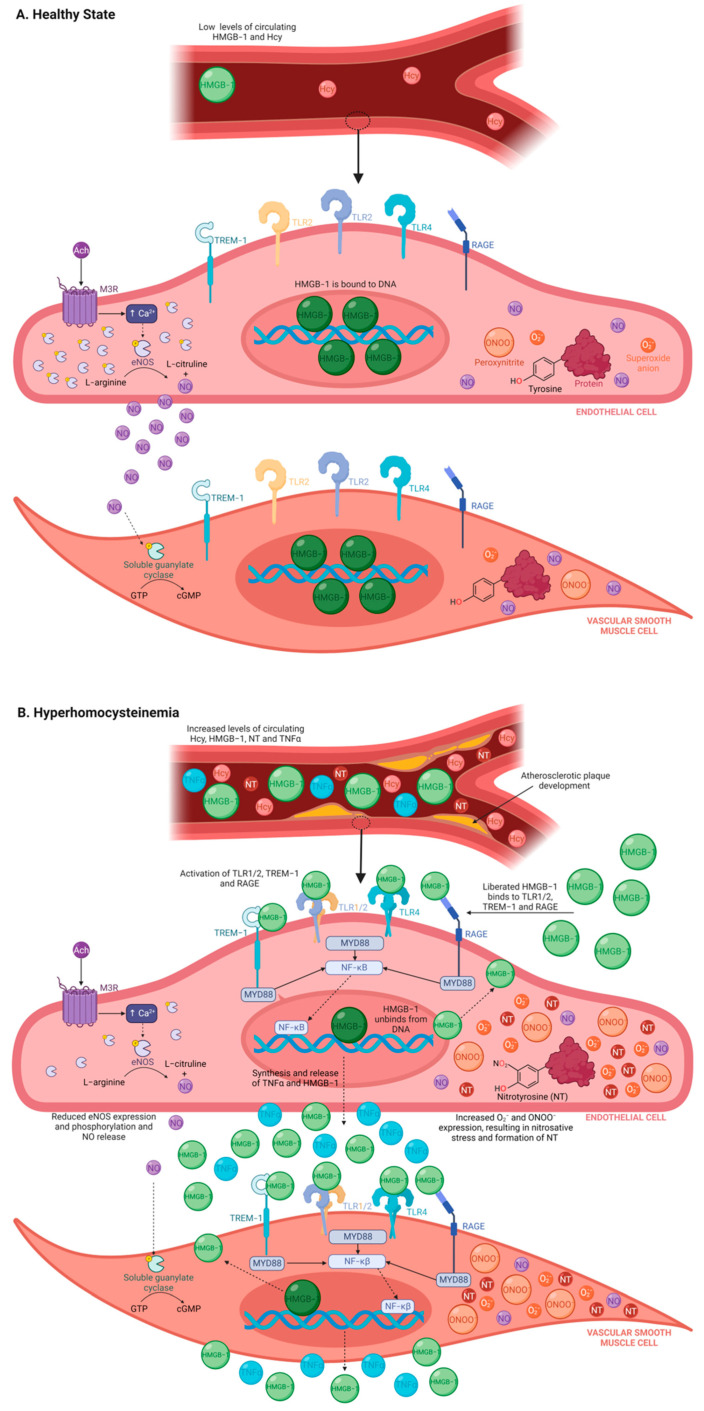
Schematic diagram of the role of HMGB-1 during healthy state and HHcy. (**A**) In a healthy state, systemic levels of HMGB-1 and Hcy are low. This is due to (i) correct metabolism, conversion and recycling of Hcy during methionine metabolism, and (ii) HMGB-1 remaining bound to DNA and thus is prevented from being liberated (either by passive release or endocytosis) into the extracellular milieu where it can bind to immune receptors (i.e., TLR1/2, TLR4, TREM-1 and RAGE) or where it behaves as a pro-inflammatory cytokine. Moreover, during normal vascular function, Ach binds to the muscarinic 3 receptor, which causes phosphorylate eNOS to convert L-arginine into L-citrulline and NO. NO then diffuses into the vascular smooth muscle cell, which triggers soluble guanylate cyclase to convert guanosine triphosphate to cyclic guanosine monophosphate, inducing relaxation of vascular smooth muscle cells. (**B**) HHcy is defined as an abnormally elevated systemic level of Hcy, which triggers oxidative stress and inflammatory pathways. Additionally, eNOS expression, phosphorylation and activity are reduced, resulting in decreased NO production and diminished relaxation. Furthermore, HMGB-1 becomes unbound from DNA and is released by the cell into the extracellular environment where it activates immune receptors and ultimately causes synthesis and release of pro-inflammatory cytokines (i.e., TNFα and HMGB-1). HHcy also causes nitrosative stress, which involves tyrosine residues being converted to nitrotyrosine by peroxynitrite (created from superoxide anions and NO). As a result, there is an increase in systemic release of nitrotyrosine and TNFα and development of atherosclerotic plaques. Abbreviations: Ach, acetylcholine; Ca^2+^, calcium; cGMP, cyclic guanosine monophosphate; eNOS, endothelial nitric oxide synthase; GTP, guanosine triphosphate; Hcy, homocysteine; HHcy, Hyperhomocysteinemia; HMGB-1, high-mobility group box-1; MyD88, myeloid differentiation actor 88; M3R, muscarinic 3 receptor; NF-KB, nuclear factor-kappa B; NO, nitric oxide; NT, nitrotyrosine; ONOO^−^, peroxynitrite; O_2_^−^, superoxide anion; RAGE, receptor for advanced glycated end-products; TLR, toll-like receptor; TREM-1, triggering receptor expressed on myeloid cells-1. Figure made using Biorender.com.

**Figure 2 nutrients-15-03186-f002:**
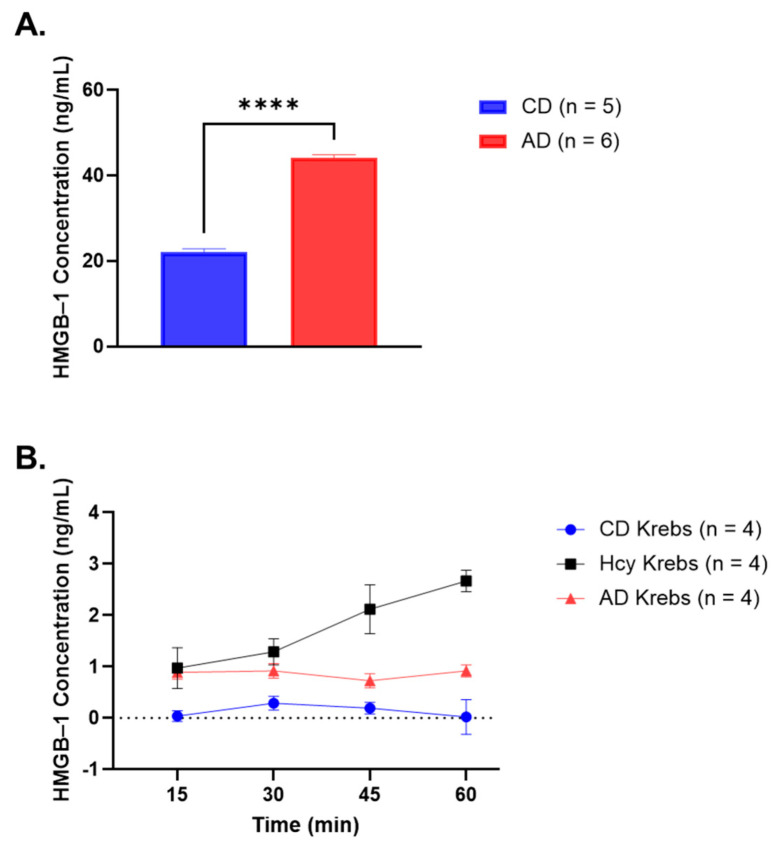
HMGB-1 concentrations in rabbit serum and Krebs taken from ex vivo organ bath studies. (**A**) Serum HMGB-1 was significantly increased in rabbits fed a 4-week AD when compared to rabbits fed a CD (mean ± SEM is shown, ***** p* < 0.0001). (**B**) Krebs collected from organ baths containing AA rings incubated with Hcy had significantly increased HMGB-1 concentration at all time points (mean ± SEM, significance shown in Table 1) when compared to Krebs from organ baths containing CD AA. Krebs from AD AA rings had elevated HMGB-1 levels at 15 min and 60 min time points (mean ± SEM, significance shown in Table 1). AA rings incubated with Hcy had significantly increased HMGB-1 concentration 45 min and 60 min when compared to AD Krebs (mean ± SEM, significance shown in Table 1). Abbreviations: AA, abdominal aorta; AD, atherogenic diet; CD, standard control diet; Hcy, homocysteine; HMGB-1, high-mobility group box-1; Krebs, Krebs–Henseleit solution; SEM, standard error of mean.

**Figure 3 nutrients-15-03186-f003:**
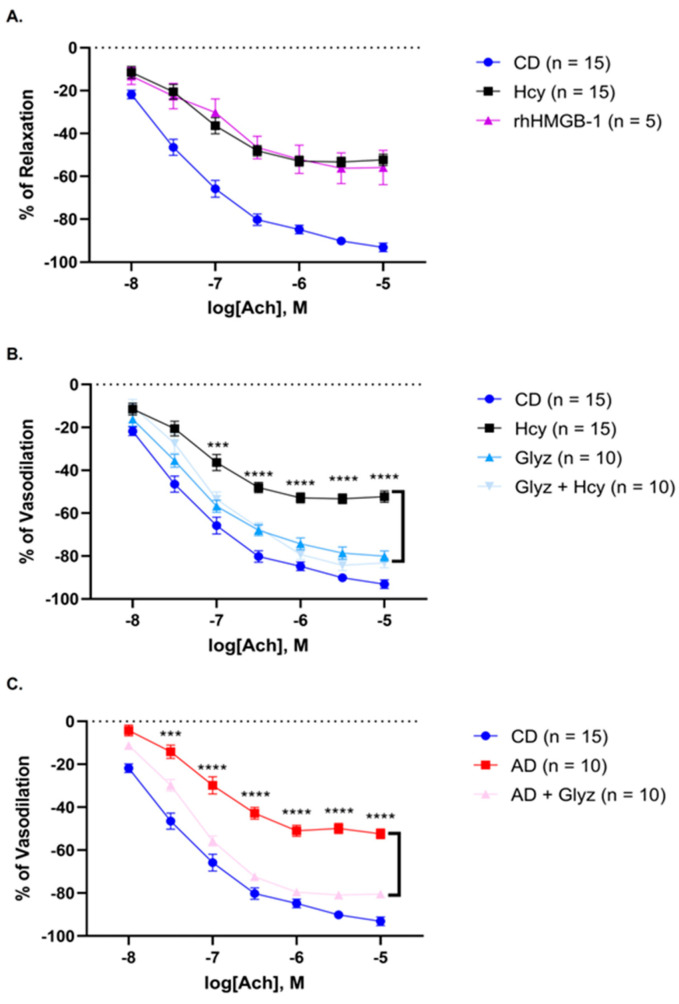
Rabbit AA relaxation responses to Ach in different diet and treatment groups. (**A**) Exposure to Hcy or rhMGB-1 resulted in a markedly similar reduced relaxation to Ach, when compared to untreated CD AA rings (mean ± SEM is shown, significance presented in Table 2). (**B**) Pre-treatment with Glyz to inhibit HMGB-1 significantly enhanced relaxation in response to Ach (mean ± SEM is shown, *** *p* < 0.001, **** *p* < 0.0001); however, Glyz treatment in the absence of Hcy resulted in reduced relaxation (mean ± SEM is shown, significance presented in Table 3). (**C**) AA rings from AD rabbits had significantly reduced relaxation in response to Ach (mean ± SEM is shown, significance presented in Table 4) which was greatly enhanced after treatment with Glyz (mean ± SEM is shown, *** *p* < 0.001, **** *p* < 0.0001). Abbreviations: AA, abdominal aorta; Ach, acetylcholine; AD, atherogenic diet; CD, standard control chow diet; Hcy, homocysteine; HMGB-1, high-mobility group box-1; Glyz, Glycyrrhizic acid; rhHMGB-1, recombinant human high-mobility group box-1; SEM, standard error of mean.

**Figure 4 nutrients-15-03186-f004:**
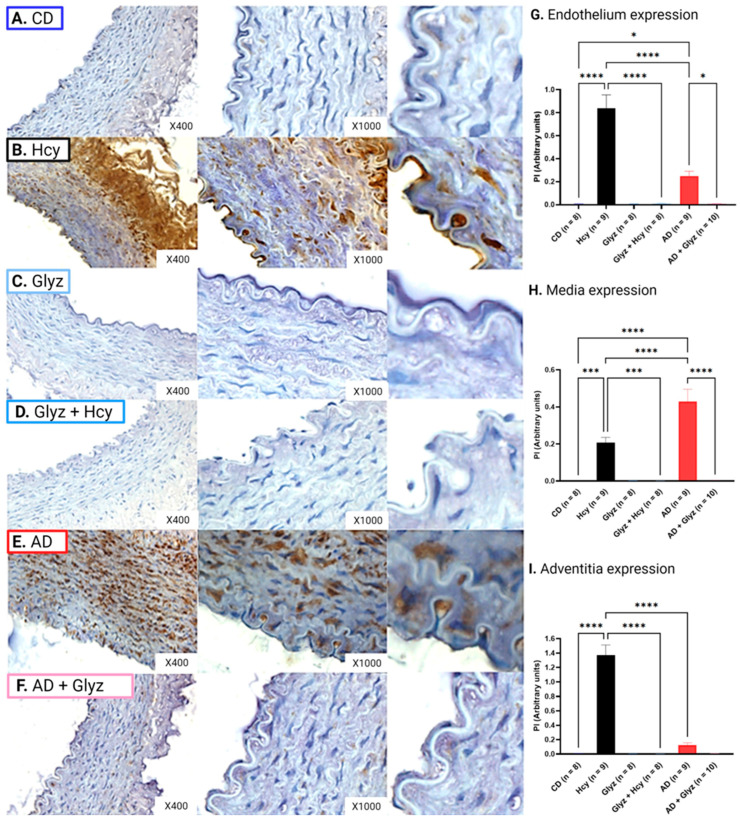
HMGB-1 expression in different layers of rabbit AA. Incubation with Hcy (**B**) resulted in significantly elevated levels of HMGB-1 expression within the endothelium (**G**) (mean ± SEM, *p* < 0.0001), media (**H**) (mean ± SEM, *p* < 0.0001) and adventitia (**I**) (mean ± SEM, *p* < 0.0001) when compared to rings from CD rabbits (**A**). Pre-treatment with Glyz (**D**) was prevented increased expression of HMGB-1 triggered by Hcy incubation within the endothelium (**G**) (mean ± SEM, *p* < 0.0001), media (**H**) (mean ± SEM, *p* < 0.0001) and adventitia (**I**) (mean ± SEM, *p* < 0.0001), when compared to Hcy rings. A 4-week AD (**E**) also resulted in increased expression of HMGB1 within the endothelium (**G**) (mean ± SEM, *p* < 0.05) and media (**H**) (mean ± SEM, *p* < 0.0001) but not in the adventitia (**I**) when compared to CD rings (**A**). Glyz was able to block increased expression of HMGB-1 after a 4-week AD (**F**) when compared to the endothelium (**G**) (mean ± SEM, *p* < 0.05) and media (**H**) (mean ± SEM, *p* < 0.0001) of AD rings. Please note that to better visualize the endothelium for the image, a section of the endothelium was cropped and enlarged from each group ×1000 image, no quantification was done on these images. Abbreviations: AA, abdominal aorta; AD, atherogenic diet; CD, standard control diet; Hcy, homocysteine; HMGB-1, high-mobility group box-1; Glyz, Glycyrrhizic acid; SEM, standard error of mean. Key: * = *p* < 0.05; *** = *p* < 0.001; **** = *p* < 0.0001; dark blue = control; black = Hcy; light blue = Glyz; medium blue = Hcy + Glyz; red = AD and light pink = AD + Glyz.

**Figure 5 nutrients-15-03186-f005:**
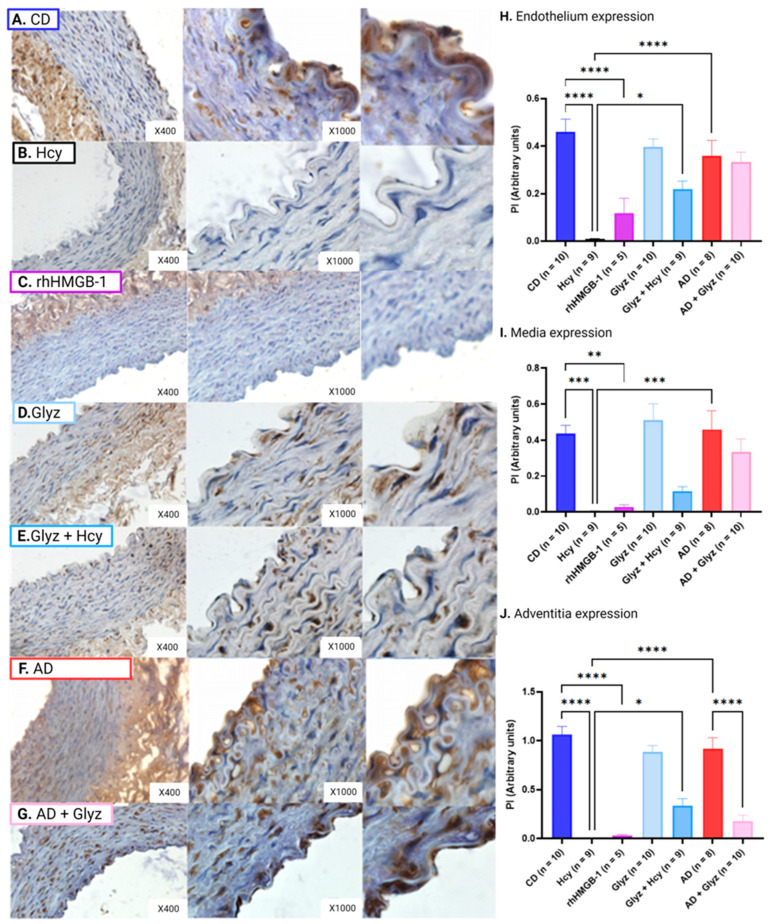
eNOS expression in different layers of rabbit AA. AA rings from CD rabbits (**A**) had high levels of eNOS expression in all blood vessel layers, which was drastically reduced in the endothelium (**H**) (mean ± SEM, *p* < 0.0001), media (**I**) (mean ± SEM, *p* < 0.0001, *p* < 0.001) and adventitia (**J**) (mean ± SEM, *p* < 0.0001) after exposure to Hcy or rhHMGB-1. Pre-treatment with Glyz prior to Hcy incubation (**D**) was able to markedly increase eNOS expression in the endothelium (**G**) (mean ± SEM, *p* < 0.05) and adventitia (mean ± SEM, *p* < 0.0001). Interestingly, no differences in eNOS expression were observed between 4-week AD and CD. Moreover, AD rings treated with Glyz (**F**) did not increase eNOS expression within the endothelium or media but did reduce eNOS expression in the adventitia (mean ± SEM, *p* < 0.0001). Please note that to better visualize the endothelium for the image, a section of the endothelium was cropped and enlarged from each group ×1000 image, no quantification was done on these images. Abbreviations: AA, abdominal aorta; Ach, acetylcholine; AD, atherogenic diet; CD, standard control diet; Hcy, homocysteine; HMGB-1, high-mobility group box-1; Glyz, Glycyrrhizic acid; rhHMGB-1, recombinant human high-mobility group box-1; SEM, standard error of mean. Key: * = *p* < 0.05; ** = *p* < 0.01; *** = *p* < 0.001; **** = *p* < 0.0001; dark blue = control; black = Hcy; purple = rhHMGB-1; light blue = Glyz; medium blue = Hcy + Glyz; red = AD and light pink = AD + Glyz.

**Figure 6 nutrients-15-03186-f006:**
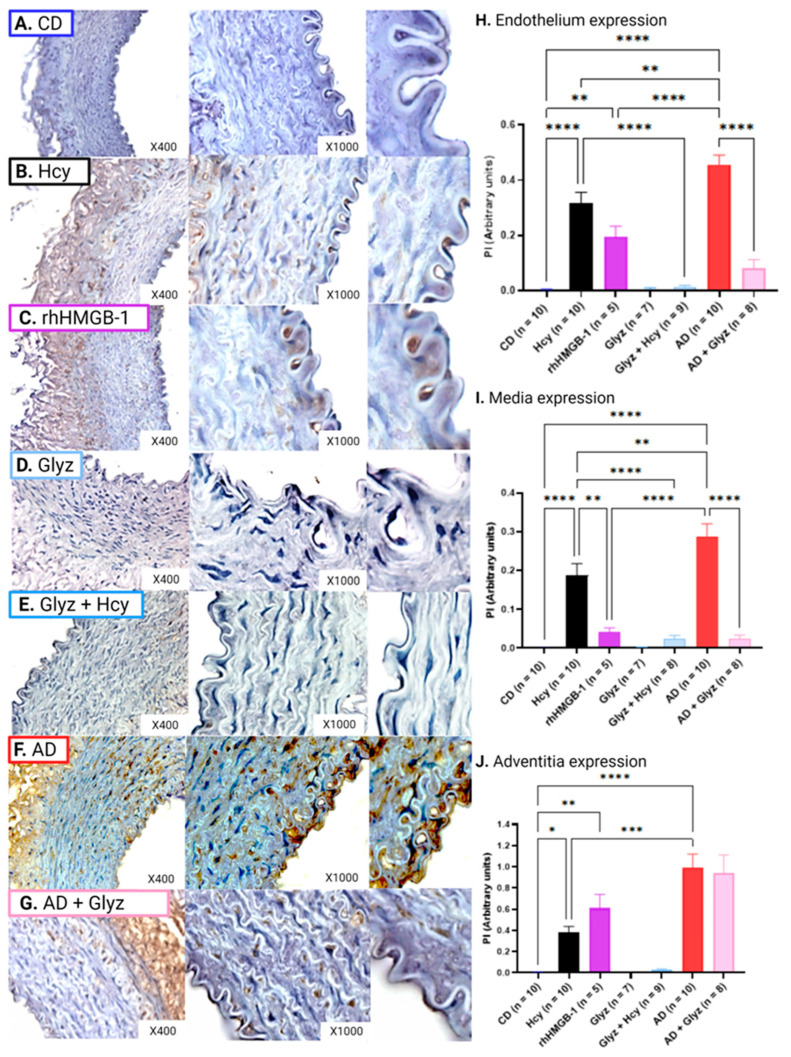
Nitrotyrosine expression in different layers of rabbit AA. Incubation with Hcy (**B**) or rhHMGB-1 (**C**) resulted in increased peptide expression of nitrotyrosine within the endothelium (**H**) (mean ± SEM, *p* < 0.0001, *p* < 0.01), media (**I**) (mean ± SEM, *p* < 0.0001) and adventitia (**J**) (mean ± SEM, *p* < 0.05, *p* < 0.01) when compared to CD rings (**A**). Again, Glyz treatment before Hcy exposure (**E**) was able to block increased nitrotyrosine expression in the endothelium (**H**) (mean ± SEM, *p* < 0.0001) and media (**I**). When compared to CD rings, AA rings from AD (**F**) had significantly elevated levels of nitrotyrosine in the endothelium (**H**) (mean ± SEM, *p* < 0.0001), media (mean ± SEM, *p* < 0.0001) and adventitia (mean ± SEM, *p* < 0.0001). AD rings treated with Glyz (**G**) showed a marked reduction in nitrotyrosine expression in the endothelium (**H**) (mean ± SEM, *p* < 0.0001) and media (**I**) (mean ± SEM, *p* < 0.0001) but not in the adventitia (**J**). Please note that to better visualize the endothelium for the image, a section of the endothelium was cropped and enlarged from each group ×1000 image, no quantification was done on these images. Abbreviations: AA, abdominal aorta; Ach, acetylcholine; AD, atherogenic diet; CD, standard control diet; Hcy, homocysteine; HMGB-1, high-mobility group box-1; Glyz, Glycyrrhizic acid; rhHMGB-1, recombinant human high-mobility group box-1; SEM, standard error of mean. Key: * = *p* < 0.05; ** = *p* < 0.01; *** = *p* < 0.001; **** = *p* < 0.0001; dark blue = control; black = Hcy; purple = rhHMGB-1; light blue = Glyz; medium blue = Hcy + Glyz; red = AD and light pink = AD + Glyz.

**Figure 7 nutrients-15-03186-f007:**
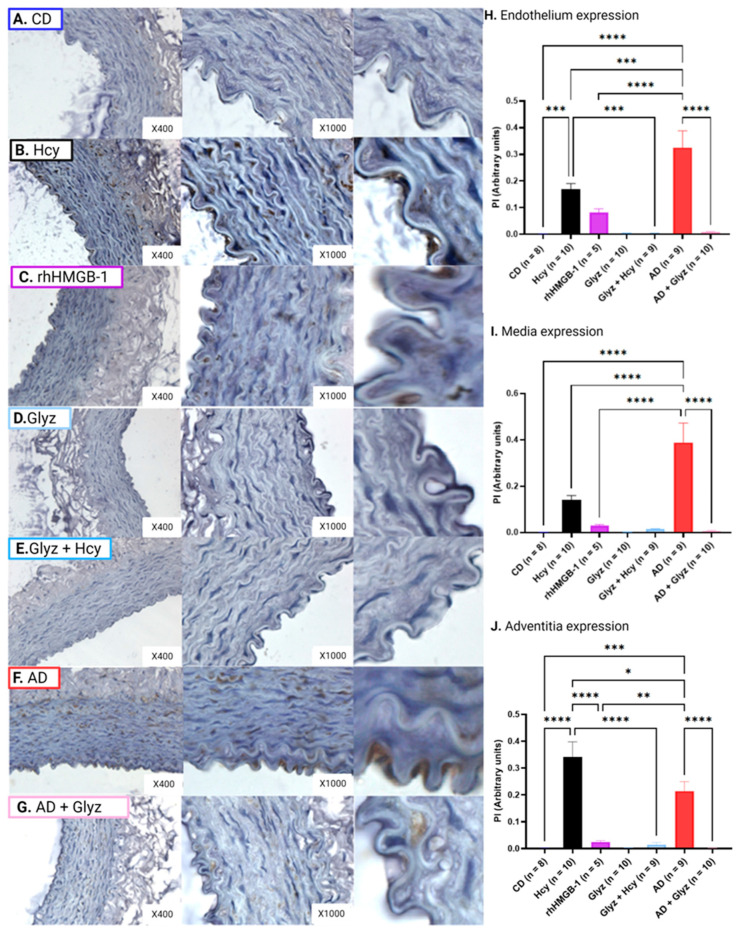
TNFα expression in different layers of rabbit AA. TNFα expression was significantly upregulated in the media (**I**) (mean ± SEM, *p* < 0.001) and adventitia (**J**) (mean ± SEM, *p* < 0.0001) after Hcy (**B**) exposure when compared to CD rings (**A**). Glyz treatment prior to Hcy (**D**) incubation normalized TNFα levels within the media (**I**) (mean ± SEM, *p* < 0.001) and adventitia (**J**) (mean ± SEM, *p* < 0.0001). Similarly, a 4-week AD significantly enhanced TNFα protein expression within the endothelium (**G**) (mean ± SEM, *p* < 0.0001), media (**I**) (mean ± SEM, *p* < 0.001) and adventitia (mean ± SEM, *p* < 0.0001), which was reduced by Glyz (**F**) (mean ± SEM, *p* < 0.0001). Please note that to better visualize the endothelium for the image, a section of the endothelium was cropped and enlarged from each group ×1000 image, no quantification was done on these images. Abbreviations: AA, abdominal aorta; Ach, acetylcholine; AD, atherogenic diet; CD, standard control diet; Hcy, homocysteine; HMGB-1, high-mobility group box-1; Glyz, Glycyrrhizic acid; rhHMGB-1, recombinant human high-mobility group box-1; TNFα; tumor necrosis factor alpha; SEM, standard error of mean. Key: * = *p* < 0.05; ** = *p* < 0.01; *** = *p* < 0.001; **** = *p* < 0.0001; dark blue = control; black = Hcy; purple = rhHMGB-1; light blue = Glyz; medium blue = Hcy + Glyz; red = AD and light pink = AD + Glyz.

**Figure 8 nutrients-15-03186-f008:**
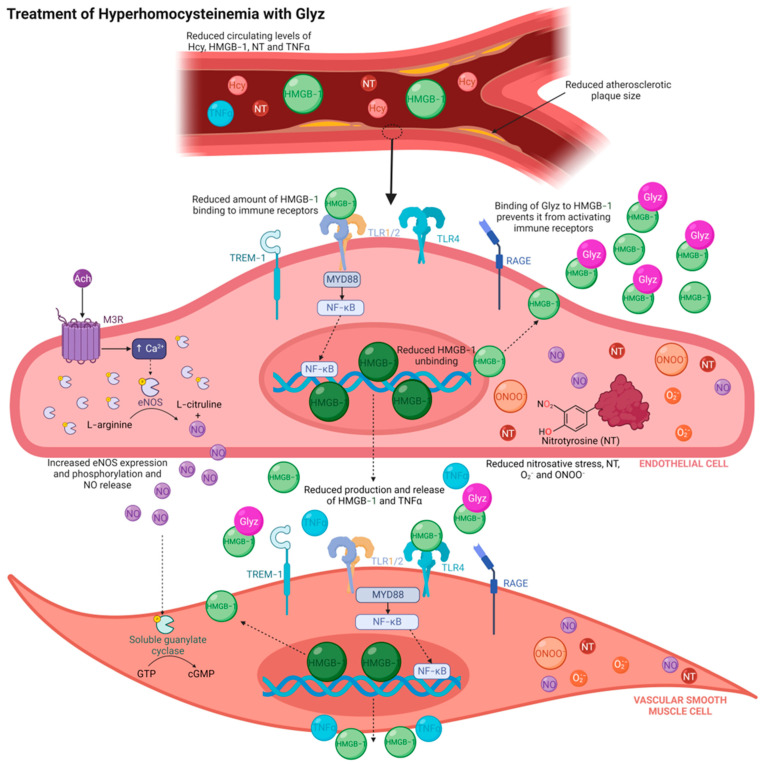
Proposed mechanism of Glyz treatment during HHcy. Treating HHcy with Glyz may reduce pathology by (i) preventing HMGB-1 from becoming unbound from DNA; (ii) decreasing HMGB-1 from activating immune receptors and subsequent inflammatory mechanisms and release of pro-inflammatory cytokines (i.e., TNFα); (iii) reducing nitrosative stress and nitrotyrosine expression; and (iv) increasing eNOS peptide expression and activity. Abbreviations: Ach, acetylcholine; Ca^2+^, calcium; cGMP, cyclic guanosine monophosphate; eNOS, endothelial nitric oxide synthase; Glyz, glycyrrhizic acid; GTP, guanosine triphosphate; Hcy, homocysteine; HHcy, Hyperhomocysteinemia; HMGB-1, high-mobility group box-1; MyD88, myeloid differentiation actor 88; M3R, muscarinic 3 receptor; NF-KB, nuclear factor-kappa B; NO, nitric oxide; NT, nitrotyrosine; ONOO^−^, peroxynitrite; O_2_^−^, superoxide anion; RAGE, receptor for advanced glycated end-products; TLR, toll-like receptor; TREM-1, triggering receptor expressed on myeloid cells-1. Figure made using Biorender.com.

**Table 1 nutrients-15-03186-t001:** Significance in concentration of HMGB-1 secretion in tested Krebs from organ baths between groups at different time intervals, obtained from Figure 2B.

Time Intervals (min)	CD Krebs vs. AD Krebs (ng/mL) (Mean ± SEM)	CD Krebs vs. Hcy Krebs (ng/mL) (Mean ± SEM)	Hcy Krebs vs. AD Krebs (ng/mL) (Mean ± SEM)
15	CD: 0.42 ± 0.10 vs. AD: 0.89 ± 0.13, ***p* = 0.0496**	CD: 0.42 ± 0.10 vs. Hcy: 0.97 ± 0.40, ***p*** **= 0.0278**	Hcy: 0.97 ± 0.40 vs. AD: 0.89 ± 0.13, *p* = 0.9705
30	CD: 0.29 ± 0.13 vs. AD: 0.92 ± 0.14, *p* = 0.1791	CD: 0.29 ± 0.13 vs. Hcy: 1.29 ± 0.25, ***p*** **= 0.0178**	Hcy: 1.29 ± 0.25 vs. AD: 0.92 ± 0.14, *p* = 0.411
45	CD: 0.19 ± 0.11 vs. AD: 0.73 ± 0.11, *p* = 0.2808	CD: 0.19 ± 0.11 vs. Hcy: 2.12 ± 0.48, ***p*** **< 0.0001**	Hcy: 2.12 ± 0.48 vs. AD: 0.73 ± 0.11, ***p*** **= 0.0008**
60	CD: 0.02 ± 0.34 vs. AD: 0.92 ± 0.12, ***p* = 0.0336**	CD: 0.02 ± 0.34 vs. Hcy: 2.67 ± 0.21, ***p*** **< 0.0001**	Hcy: 2.67 ± 0.21 vs. AD: 0.92 ± 0.12, ***p*** **< 0.0001**

Abbreviations: AD, atherogenic diet; CD, standard control chow diet; Hcy, homocysteine; Krebs, Krebs–Henseleit solution; SEM, standard error of the mean. Bolded *p* value represents significance between groups being compared.

**Table 2 nutrients-15-03186-t002:** Significance of relaxation responses between groups at different Ach concentrations, obtained from Figure 3A.

Log [Ach], M	CD vs. Hcy (%) (Mean ± SEM)	CD vs. rhHMGB-1 (%) (Mean ± SEM)	Hcy vs. rhHMGB-1 (%) (Mean ± SEM)
−8.0	CD: −21.83 ± 2.04 vs. Hcy: −11.43 ± 2.75, ***p* = 0.0319**	CD: −21.83 ± 2.04 vs. rhHMGB-1: −13.20 ± 3.92, *p* = 0.2989	Hcy: −11.43 ± 2.75 vs. rhHMGB-1: −13.20 ± 3.92, *p* = 0.9500
−7.5	CD: −46.48 ± 3.76 vs. Hcy: −20.58 ± 3.49, ***p*** **< 0.0001**	CD: −46.48 ± 3.76 vs. rhHMGB-1: −22.53 ± 5.93, ***p* = 0.0020**	Hcy: −20.58 ± 3.49 vs. rhHMGB-1: −22.53 ± 5.93, *p* = 0.9398
−7.0	CD: −65.75 ± 3.99 vs. Hcy: −36.37 ± 3.75, ***p*** **< 0.0001**	CD: −46.48 ± 3.76 vs. rhHMGB-1: −30.17 ± 6.37, ***p*** **< 0.0001**	Hcy: −36.37 ± 3.75 vs. rhHMGB-1: −30.17 ± 6.37, *p* = 0.5*346*
−6.5	CD: −80.21 ± 2.65 vs. Hcy: −48.06 ± 2.47, ***p*** **< 0.0001**	CD: −46.48 ± 3.76 vs. rhHMGB-1: −46.53 ± 5.27, ***p*** **< 0.0001**	Hcy: −48.06 ± 2.47 vs. rhHMGB-1: −46.53 ± 5.27, *p* = 0.9625
−6.0	CD: −84.80 ± 2.00 vs. Hcy: −52.87 ± 2.31, ***p*** **< 0.0001**	CD: −46.48 ± 3.76 vs. rhHMGB-1: −52.04 ± 6.55, ***p*** **< 0.0001**	Hcy: −52.87 ± 2.31 vs. rhHMGB-1: −52.04 ± 6.55, *p* = 0.9890
−5.5	CD: −90.10 ± 1.32 vs. Hcy: −53.30 ± 2.21, ***p*** **< 0.0001**	CD: −46.48 ± 3.76 vs. rhHMGB-1: −56.18 ± 7.18, ***p*** **< 0.0001**	Hcy: −53.30 ± 2.21 vs. rhHMGB-1: −56.18 ± 7.18, *p* = 0.8731
−5.0	CD: −93.13 ± 1.97 vs. Hcy: −52.29 ± 2.71, ***p*** **< 0.0001**	CD: −46.48 ± 3.76 vs. rhHMGB-1: −55.84 ± 8.02, ***p*** **< 0.0001**	Hcy: −52.29 ± 2.71 vs. rhHMGB-1: −55.84 ± 8.02, *p* = 0.8116

Abbreviations: Ach, acetylcholine; CD, standard control chow diet; Hcy, homocysteine; rhHMGB-1, recombinant human high-mobility group box-1; SEM, standard error of mean. Bolded *p* value represents significance between groups being compared.

**Table 3 nutrients-15-03186-t003:** Significance of relaxation responses between groups at different Ach concentrations, obtained from Figure 3B.

Log [Ach], M	CD vs. Hcy (%) (Mean ± SEM)	CD vs. Glyz (%) (Mean ± SEM)	Hcy vs. Glyz + Hcy (%) (Mean ± SEM)
−8.0	CD: −21.83 ± 2.04 vs. Hcy: −11.43 ± 2.75, ***p* = 0.0319**	CD: −21.83 ± 2.04 vs. Glyz: −16.20 ± 3.14, *p* = 0.5321	Hcy: −11.43 ± 2.75 vs. Glyz + Hcy: −10.28 ± 3.35, *p* = 0.9928
−7.5	CD: −46.48 ± 3.76 vs. Hcy: −20.58 ± 3.49, ***p*** **< 0.0001**	CD: −46.48 ± 3.76 vs. Glyz: −35.51 ± 3.00, ***p* = 0.0444**	Hcy: −20.58 ± 3.49 vs. Glyz + Hcy: −27.45 ± 4.45, *p* = 0.3550
−7.0	CD: −65.75 ± 3.99 vs. Hcy: −36.37 ± 3.75, ***p*** **< 0.0001**	CD: −65.75 ± 3.99 vs. Glyz: −56.79 ± 2.86, *p* = 0.1403	Hcy: −36.37 ± 3.75 vs. Glyz + Hcy: −53.56 ± 3.42, ***p* = 0.0003**
−6.5	CD: −80.21 ± 2.65 vs. Hcy: −48.06 ± 2.47, ***p*** **< 0.0001**	CD: −80.21 ± 2.65 vs. Glyz: −67.96 ± 2.48, ***p*** **= 0.0186**	Hcy: −48.06 ± 2.47 vs. Glyz + Hcy: −66.87 ± 2.49, ***p* < 0.0001**
−6.0	CD: −84.80 ± 2.00 vs. Hcy: −52.87 ± 2.31, ***p*** **< 0.0001**	CD: −84.80 ± 2.00 vs. Glyz: −74.32 ± 2.80, *p* = 0.0602	Hcy: −52.87 ± 2.31 vs. Glyz + Hcy: −79.26 ± 2.33, ***p* < 0.0001**
−5.5	CD: −90.10 ± 1.32 vs. Hcy: −53.30 ± 2.21, ***p*** **< 0.0001**	CD: −84.80 ± 2.00 vs. Glyz: −78.64 ± 2.83, ***p*** **= 0.0322**	Hcy: −53.30 ± 2.21 vs. Glyz + Hcy: −84.26 ± 2.53, ***p* < 0.0001**
−5.0	CD: −93.13 ± 1.97 vs. Hcy: −52.29 ± 2.71, ***p*** **< 0.0001**	CD: −93.13 ± 1.97 vs. Glyz: −80.01 ± 2.48, ***p*** **= 0.0105**	Hcy: −52.29 ± 2.71 vs. Glyz + Hcy: −83.20 ± 2.28, ***p* < 0.0001**

Abbreviations: Ach, acetylcholine; CD, standard control chow diet; Hcy, homocysteine; Glyz, Glycyrrhizic acid; SEM, standard error of mean. Bolded *p* value represents significance between groups being compared.

**Table 4 nutrients-15-03186-t004:** Significance of relaxation responses between groups at different Ach concentrations, obtained from Figure 3C.

Log [Ach], M	CD vs. AD (%) (Mean ± SEM)	AD vs. AD + Glyz (%) (Mean ± SEM)
−8.0	CD: −21.83 ± 2.04 vs. AD: −4.21 ± 2.48, ***p* < 0.0001**	AD: −4.21 ± 2.48 vs. AD + Glyz: −11.21 ± 1.60, *p* = 0.1850
−7.5	CD: −46.48 ± 3.76 vs. AD: −14.15 ± 3.07, ***p*** **< 0.0001**	AD: −14.15 ± 3.07 vs. AD + Glyz: −29.711 ± 2.66, ***p* = 0.0040**
−7.0	CD: −65.75 ± 3.99 vs. AD: −29.83 ± 4.06, ***p*** **< 0.0001**	AD: −29.83 ± 4.06 vs. AD + Glyz: −55.72 ± 2.35, ***p*** **< 0.0001**
−6.5	CD: −80.21 ± 2.65 vs. AD: −42.77 ± 2.78, ***p*** **< 0.0001**	AD: −42.77 ± 2.78 vs. AD + Glyz: −72.39 ± 1.01, ***p*** **< 0.0001**
−6.0	CD: −84.80 ± 2.00 vs. AD: −50.95 ± 2.62, ***p*** **< 0.0001**	AD: −50.95 ± 2.62 vs. AD + Glyz: −79.56 ± 1.13, ***p*** **< 0.0001**
−5.5	CD: −90.10 ± 1.32 vs. AD: −49.91 ± 2.38, ***p*** **< 0.0001**	AD: −49.91 ± 2.38 vs. AD + Glyz: −80.85 ± 1.19, ***p*** **< 0.0001**
−5.0	CD: −93.13 ± 1.97 vs. AD: −52.35 ± 2.23, ***p*** **< 0.0001**	AD: −52.35 ± 2.23 vs. AD + Glyz: −80.42 ± 1.46, ***p*** **< 0.0001**

Abbreviations: Ach, acetylcholine; AD, atherogenic diet; CD, standard control chow diet; Hcy, homocysteine; Glyz, Glycyrrhizic acid; SEM, standard error of mean. Bolded *p* value represents significance between groups being compared.

## Data Availability

The data presented in this study are available on request from the corresponding author.

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
