# Peer review of "Glycyrrhizic Acid Inhibits High-Mobility Group Box-1 and Homocysteine-Induced Vascular Dysfunction"

_nutrients, 2023, doi:10.3390/nu15143186_

Round 1
Reviewer 1 Report
The article titled "Glycyrrhizic acid inhibits high-mobility group box-1 and homocysteine-induced vascular dysfunction" addresses an interesting issue regarding the role of HMGB-1 during acute HHcy and HHcy-Induced atherogenesis, in an animal model, as well as the potential inhibitory effect of glycyrrhizic acid (Glyz) on vascular function.
The manuscript has several problems that need to be addressed:
Introduction: the purpose should be described in more detail and should reflect the title of the article. The title is focused on GlyZ, instead in lines 85-95 the authors speak only of HMGB-1. Please edit and integrate.
Methods: The authors would like to evaluate the effect of Glyz as an inhibitor of HMGB-1 in homocysteine-induced vascular dysfunction. For this purpose, they use an animal model represented by rabbits fed with a normal diet (n=25) or with an atherogenic diet (n=10). The authors should also indicate the number of animals they have allocated to the different treatments and measurements carried out, considering that in some cases they have indicated that they have used a different number from the initial one.
Line 150: krebs samples. What is meant by krebs samples?
Line 155: organ bath. What is meant by organ bath? The authors actually do an ex vivo study by incubating abdominal aortic (AA) rings in a Krebs buffer and analyzing the release of molecules under these conditions for up to one hour. This method must be reported more clearly and the reason for the method used and its purposes must also be explained.
Line 178: 3mM Hcy: why such a high concentration of HCY? Have you tried more physiological concentrations than are found in vivo?
Statistic: Explain why they used SEM instead of standard deviation. In my opinion, since these are experimental data from a limited sample, the data should be presented as mean and standard deviation.
Results: Results should be better presented and described:
Fig. 1a, why did they use so few animals? ( n=4 for CD and n=5 for AD?. ) It would be advisable to report the data of hmgb as mean and DS on all the animals analyzed or normal diet ( n= 25) or with an atherogenic diet (n = 10).
In the tables, the authors indicate the p-value without indicating the value. In my opinion, all tables should be modified by presenting data as mean and standard deviation and their associated significances.
Line 224: Krebs samples: replace here and thereafter the term Krebs samples with the tested samples or AA samples or supernatants of AA samples or better AA rings.
Line 229: AA rings incubated with Hcy had significantly increased HMGB-1 concentration 45 min and 60 min when compared to AD Krebs. What is the significance of this experiment? The study hypothesis is that 1hr of treatment with Hcy could induce an increased expression of HMGB-1 or an increased release? And What happens in the absence of Hcy ?
Line 232: why sem? Replace sem with DS. In fact, the standard deviation indicates the variability of a measurement made on the sample. Standard deviation (SD) is used to understand how "distributed" a data set is. The standard error (SE) or standard error of the mean (SEM) is used to estimate the mean of a population.
Line 264 and table 1: modify the table by inserting not only the significances but also the values, the SDs.
ine 237: What is the meaning of this experiment? the authors used an atherogenic diet rich in methionine a precursor of hcy. What are the levels of hcy in the animal serum with normal and atherogenic diet?
Why do the authors add hcy ex vivo? Please explain.
Table 2: and table 3: modify the tables by inserting not only the significances but also the values, the DS .
Figures 3-6 need to be presented better.
In the immunohistochemistry figures, the authors should add the quantification of the total expression of the studied marker in the different treatments (control, Hcy, etc) and better present that at the different levels (endothelium, etc) . Moreover, they should also explain how identify the different levels ( intima, etc..).
In general, the authors want to study the effect of Glyz on vascular function as AA rings in the presence of acute HCy treatment (3 mM) and in the presence of chronic treatment with HCY such as that obtained with an atherogenic diet. I have two comments. Why was such a high concentration of ex vivo HCY used? The authors tested what happens at different concentrations of HCY in the presence or absence of inflammatory stimuli such as TNF?. In fact, previous works have shown that HCY enhances the pro-inflammatory action of TNF on endothelial activation which is counteracted by the action of natural polyphenols such as resveratrol and hydroxytyrosol in a concentration-dependent manner (doi: 10.1152/ajpheart.00432.2007). To characterize the effect of a compound and its mechanism of action it is essential to analyze its dose response. In fact, different doses of the same compound can lead to even opposite effects . The authors should add data regarding the dose response of Gly in an ex vivo model of AA rings treated with different Hcy doses.
Finally, the authors should describe in a more detailed way the rational bases of the Glyz as an inhibitor of HMGB-1.
Author Response
Reviewer 1
The article titled "Glycyrrhizic acid inhibits high-mobility group box-1 and homocysteine-induced vascular dysfunction" addresses an interesting issue regarding the role of HMGB-1 during acute HHcy and HHcy-Induced atherogenesis, in an animal model, as well as the potential inhibitory effect of glycyrrhizic acid (Glyz) on vascular function.
The manuscript has several problems that need to be addressed:
Comment #1: Introduction – the purpose should be described in more detail and should reflect the title of the article. The title is focused on GlyZ, instead in lines 85-95 the authors speak only of HMGB-1. Please edit and integrate.
We greatly appreciate this comment and agree that omitting Glyz from the aims misleads the reader to the purpose of the study. We have rewritten the aims to include Glyz in the aims and have added a sentence that situates the reader with what is known about Glyz in CVD experimental and clinical studies. The sentence added to better situate the reader reads as:
Line 83-86: Results from experimental and clinical studies have reported cardioprotective effects of Glyz, including anti-inflammatory, antioxidant, anti-thrombotic and anti-atherosclerotic [23]; however, its role during Hcy-induced pathology remains poorly understood.
The aims have been rewritten and now read as:
Line 112-124: The purpose of the present study was to further elucidate the pathogenic role that HMGB-1 has during acute and chronic HHcy-induced vascular dysfunction and to evaluate Glyz (HMGB-1 inhibitor) as a potential therapeutic option to restore normal vascular function. To address the aims, abdominal aortae (AA) from male New Zealand White rabbits fed either a 4-week normal chow (CD) or atherogenic diet (AD) were collected for ex vivo isometric tension myography organ bath studies and semi-quantitative immunohistochemistry analysis to investigate the effect that acute and chronic HHcy, HMGB-1 and Glyz treatment had on Ach-mediated relaxation and protein expression of markers associated with vascular dysfunction, inflammation, and cell stress (i.e., eNOS, nitrotyrosine and tumor necrosis factor alpha (TNFα). An enzyme-linked immunosorbent assay (ELSA) was also performed on serum and Krebs-Henseleit solution (Krebs) to determine the effect that acute and chronic HHcy had on systemic and isolated artery HMGB-1 release.
Comment #2: Methods – The authors would like to evaluate the effect of Glyz as an inhibitor of HMGB-1 in homocysteine-induced vascular dysfunction. For this purpose, they use an animal model represented by rabbits fed with a normal diet (n=25) or with an atherogenic diet (n=10). The authors should also indicate the number of animals they have allocated to the different treatments and measurements carried out, considering that in some cases they have indicated that they have used a different number from the initial one.
Thank you for this comment we agree that for the ease of the reader each section and figure/figure legend should detail the number of animals used. To clarify, we used a total of 25 rabbits, which were then divided into control diet (n = 15) or atherogenic diet (n = 10) once they were 12 weeks old. The following have been added to each section:
Line 141-143: Male New Zealand White rabbits (n = 25) at 8-11 weeks of age were purchased from Nanowie Small Animal Production Unit (VIC, Australia) or Flinders City University (SA, Australia), and were housed individually at the Victoria University Werribee Campus Animal Facilities.
Line 155-158: Rabbits received either a 4-week CD (n = 15) or AD (n = 10), which was supplemented with cholesterol (0.5 %) + methionine (1%) + peanut oil (5 %) (Cat#SF00-218; Specialty Feeds, VIC, AUS).
Line 178-179: To determine if a 4-week AD increases systemic release of HMGB-1, serum was collected from CD (n = 5) and AD (n = 6) rabbits.
Line 182-185: Additionally, to determine if acute or chronic HHcy alters the release of HMGB-1 from arteries ex vivo, Krebs was collected from individual organ baths containing AA rings from CD (n = 4) and AD (n = 4) rabbits and CD AA rings incubated with Hcy (n = 4) to mimic severe and acute HHcy.
Line 210-215: Rings were again re-stretched to 2 grams, refreshed, and equilibrated for a further 30 min. AA rings were then: (i) left to rest for 120 min to serve as untreated control rings (CD: n = 15; AD: n = 10); (ii) incubated with 100 nM human recombinant HMGB-1 (rhHMGB-1: n = 5) for 60 min; (iii) incubated with 3 mM Hcy (Hcy: n = 15) for 60 min to induce endothelial dysfunction observed in severe and acute HHcy; (iv) incubated with 1 µM Glyz (Glyz: n = 10; AD + Glyz: n = 10); or (v) or pre-treated with Glyz 60 min before a 60 min incubation with Hcy (Glyz + Hcy: n = 10).
Line 244-247: Due to loss of tissue during paraffin processing, n = 5-10 for immunohistochemistry studies was retrievable.
Please also note that we have added n numbers have been included in each figure.
Comment #3: Line 155: organ bath. What is meant by organ bath? The authors actually do an ex vivo study by incubating abdominal aortic (AA) rings in a Krebs buffer and analyzing the release of molecules under these conditions for up to one hour. This method must be reported more clearly and the reason for the method used and its purposes must also be explained.
Thank you for bringing this to our attention. In order to clarify to the reader why and how we measured the amount of HMGB-1 released from isolated abdominal aortic rings, we have re-written Section 2.5 of the methods:
Line 182-191: To determine if a 4-week AD increases systemic release of HMGB-1, serum was collected from CD (n = 5) and AD (n = 6) rabbits. Venous blood (10 mL) was collected at the site of inferior vena cava perforation, centrifuged at 3,500 RPM for 10 min at 4C and the separated serum was collected and stored at -80C. Additionally, to determine if acute or chronic HHcy alters the release of HMGB-1 from arteries ex vivo, Krebs was collected from individual organ baths containing AA rings from CD (n = 4) and AD (n = 4) rabbits and CD AA rings incubated with Hcy (n = 4) to mimic severe and acute HHcy. To do this, AA rings were immediately placed into organ baths (OB8, Zultek Engineering, VIC, AUS) containing 5 mL of Krebs and were maintained at 37C and continuously bubbled with 95 % carbogen (95 % oxygen; 5 % carbon dioxide) to mimic a physiologically relevant environment and were left to acclimatize to 30 min. At this time, 3 mM of Hcy was added to some rings and 300 µl was then removed from all organ baths containing CD, AD and Hcy AA rings every 15 mins over a 60 min time period and were immediately stored at -80C.
Comment #4: Line 178: 3mM Hcy: why such a high concentration of HCY? Why do the authors add hcy ex vivo? Have you tried more physiological concentrations than are found in vivo?
Thank you for this very important question. Within the literature a dose of 3mM Hcy has been used to invoke pathological states mimicking HHcy in cell culture in endothelial and vascular cells. Moreover, our laboratory has established that incubating arterial rings with 3mM Hcy is able to mimic similar dysfunction that is seen in patients with severe HHcy within 1 hour (representing an acute model of HHcy). We have now added the following to the manuscript:
Line 191-193: A 3 mM dose of Hcy was chosen for thi study as we [28,29] and others [30,31] have reported its ability to invoke pathological states mimicking severe acute HHcy in arterial organ bath and cell culture studies
Comment #5: Statistic: Explain why they used SEM instead of standard deviation. In my opinion, since these are experimental data from a limited sample, the data should be presented as mean and standard deviation.
Thank you for your suggestion. Indeed, both SEM and SD can be used, however we have consistently presented our data in all our publications as SEM, as this is what pharmacological research requires. We kindly ask if we can keep the SEM in this manuscript to maintain convention.
Comment #6: Results: Results should be better presented and described.
Thank you for bringing this to our attention as we want our manuscript to be clear and concise so that readers are able to understand our results and their interpretation. As a general edit we have now added a concluding sentence to results section to interpret what the results mean.
6a. Fig. 1a, why did they use so few animals? (n=4 for CD and n=5 for AD?. ) It would be advisable to report the data of hmgb as mean and DS on all the animals analyzed or normal diet (n= 25) or with an atherogenic diet (n = 10).
Thank you for this comment. Unfortunately, due to our rabbit suppliers reducing rate of breeding and size of breeding colonies during COVID19 and then permanently closing in 2021, acquiring rabbits was extremely difficult. Therefore, due to the inability to source rabbits in Australia and budgeting restraints we were only able to achieve small n numbers for the ELISA kit studies used to determine the amount HMGB-1 in serum and Krebs. Regarding the use of SEM over SD, please refer to Comment #5.
6b. In the tables, the authors indicate the p-value without indicating the value. In my opinion, all tables should be modified by presenting data as mean and standard deviation and their associated significances.
Thank you for this comment. We have now edited all tables and have added in the appropriate mean value ± SEM and p value.
6c. Line 224: Line 150: krebs samples. What is meant by krebs samples? Replace here and thereafter the term Krebs samples with the tested samples or AA samples or supernatants of AA samples or better AA rings.
Thank you for this comment, as it will help us to ensure that our manuscript is clear to the reader. We have since removed “Krebs samples” and have replaced with “Krebs”. The reason for this experiment was to determine if: 1. incubation with 3 mM Hcy to induce acute HHcy causes active release of HMGB-1 and 2. a 4-week atherogenic diet was able to increase HMGB-1 by arteries and compare these to normal healthy controls. Please refer to comment #3 for the rewrite of this section.
6d. Line 229: AA rings incubated with Hcy had significantly increased HMGB-1 concentration 45 min and 60 min when compared to AD Krebs. What is the significance of this experiment? The study hypothesis is that 1hr of treatment with Hcy could induce an increased expression of HMGB-1 or an increased release? And What happens in the absence of Hcy?
We greatly appreciate this comment. The reason for this experiment was to understand if Hcy incubation (to mimic acute HHcy) or a 4-week atherogenic diet (to mimic chronic HHcy) was able to increase the release of HMGB-1. We show that acute and chronic HHcy results in increased expression of HMGB-1 in Krebs and in the absence of Hcy the level of HMGB-1 release is low. We have since revised the writing of the method and results of this section aid in clarity of understanding.
6e. Line 232: why sem? Replace sem with DS. In fact, the standard deviation indicates the variability of a measurement made on the sample. Standard deviation (SD) is used to understand how "distributed" a data set is. The standard error (SE) or standard error of the mean (SEM) is used to estimate the mean of a population.
Please refer to Comment #5.
6f. Line 237: What is the meaning of this experiment? the authors used an atherogenic diet rich in methionine a precursor of hcy. What are the levels of hcy in the animal serum with normal and atherogenic diet?
Thank you for this valuable question. The reason for using a HHcy-induced atherogenic rabbit model was for us to evaluate the ability of Glyz to reduce pathology induced by Hcy in both a chronic HHcy (4-week atherogenic diet supplemented with 1% methionine) and acute HHcy (1 hr incubation with 3 mM homocysteine incubation). While not directly quantified, we have shown that a 4-week atherogenic diet causes endothelial dysfunction. We have now added the following sentence:
Line 158-163: A 4-week AD, rich in methionine (the precursor to Hcy), cholesterol and saturated fat, in leporine models has been established by our laboratory as a physiologically relevant diet to mimic a natural human diet by promoting endothelial dysfunction and sporadic lesion formation within 4 weeks [24,25] and atherosclerotic plaque development in 12 weeks [26-29]. For the purpose of this study, we are interested in the role that HMGB-1 has during endothelial dysfunction and if treatment with Glyz is beneficial.
6g. Figures 3-6 need to be presented better. In the immunohistochemistry figures, the authors should add the quantification of the total expression of the studied marker in the different treatments (control, Hcy, etc) and better present that at the different levels (endothelium, etc) . Moreover, they should also explain how identify the different levels ( intima, etc..).
Thank you this comment we have now edited Figures 4-7 (IHC results) and we have also added Supp. Figure 1, which allows the reader to locate the endothelium, media and adventitia of AA during our tracing.
Comments #7: In general, the authors want to study the effect of Glyz on vascular function as AA rings in the presence of acute HCy treatment (3 mM) and in the presence of chronic treatment with HCY such as that obtained with an atherogenic diet. I have two comments. Why was such a high concentration of ex vivo HCY used? The authors tested what happens at different concentrations of HCY in the presence or absence of inflammatory stimuli such as TNF?. In fact, previous works have shown that HCY enhances the pro-inflammatory action of TNF on endothelial activation which is counteracted by the action of natural polyphenols such as resveratrol and hydroxytyrosol in a concentration-dependent manner (doi: 10.1152/ajpheart.00432.2007). To characterize the effect of a compound and its mechanism of action it is essential to analyze its dose response. In fact, different doses of the same compound can lead to even opposite effects . The authors should add data regarding the dose response of Gly in an ex vivo model of AA rings treated with different Hcy doses.
Thank you this important suggestion. As we are unable to complete these experiments, we have added a sentence describing this as a future study.
Line 632-634: Future studies utilizing Glyz should investigate the direct effect that it has on vasoactive responses of blood vessels by performing a dose-response and its ability to alter eNOS phosphorylation and activity.
Comment #8: Finally, the authors should describe in a more detailed way the rational bases of the Glyz as an inhibitor of HMGB-1.
Thank you for this comment, we have now expanded on our section that talks about the ability of Glyz to inhibit HMGB-1 and have added original article references to support our results.
Line 582-590: Evidence from in silico studies have demonstrated the ability of Glyz to sequester and directly interact with HMGB-1 to inhibit its pathogenic function and engagement with innate immune receptors, such as TLR2/4, TREM-1 and RAGE [55,56]. Results from nuclear magnetic resonance and fluorescence studies have reported the ability of Glyz to directly associate with two shallow concave surfaces on HMGB-1 in a dose-dependent manner, resulting in the hindrance of its cytokinetic, chemoattractant and mitogen activating abilities [56,57]. Furthermore, Glyz’s pro found ability to suppress HMGB-1 release from injured cells may also be explained by its ability to inhibit HMGB-1 phosphorylation [58] and expression [59].
Reviewer 2 Report
The article presented by Gadanec et al. presents a comprehensive study on the role of the HMGB1 biomarker and the response to Glycyrrhizic Acid. The article is well written, well structured and also presents the results in a clear and orderly manner. The number of assays performed is sufficient to confirm the hypothesis proposed by the authors.
There are some issues that should be corrected by the authors before publication of the article.
-There is a lack of an explanatory figure in the introduction that puts in context the physiological process being treated and the biomolecules involved. Please include a figure that helps to better understand the biological process.
- Figure 1, on the vertical axis, should read "HMGB1" and not just concentration.
-The authors should review the exponents of the powers. In some cases it appears as 10^-5.0. Is it necessary to put 5.0 or only 5?
-On the other hand, between lines 293-302 and 334-344 the superscript has not been used. Check the whole manuscript.
-Finally, the authors do not detail in the text much information about the attached histologic images. It would be convenient to include more descriptive information about them.
As a minor detail, the name of the plant Glycyrrhiza glabra should appear in italics.
The quality of the English is correct. A proofreading by the authors themselves should be sufficient.
Author Response
Reviewer 2
The article presented by Gadanec et al. presents a comprehensive study on the role of the HMGB1 biomarker and the response to Glycyrrhizic Acid. The article is well written, well structured and also presents the results in a clear and orderly manner. The number of assays performed is sufficient to confirm the hypothesis proposed by the authors.
There are some issues that should be corrected by the authors before publication of the article.
Comment#1: There is a lack of an explanatory figure in the introduction that puts in context the physiological process being treated and the biomolecules involved. Please include a figure that helps to better understand the biological process.
Thank you for this suggestion. We have now added two figure (Figure 1A and B and Figure 8) to describe the difference in healthy and pathological states and how Glyz is working to absolve dysfunction.
Comment #2: Figure 1, on the vertical axis, should read "HMGB1" and not just concentration.
Thank you for pointing this out to us. The y-axes (vertical) on Figure 1A and B now read HMGB-1 concentration.
Comment #3: The authors should review the exponents of the powers. In some cases it appears as 10^-5.0. Is it necessary to put 5.0 or only 5? On the other hand, between lines 293-302 and 334-344 the superscript has not been used. Check the whole manuscript.
We greatly appreciate you for bringing our inconsistencies to our attention. We have now read through the entire manuscript and have made sure that superscript is used when required and that whole numbers are written with the zero omitted. For example, 10^-5.0 M is now written as 10^5 M.
Comment #4: Finally, the authors do not detail in the text much information about the attached histologic images. It would be convenient to include more descriptive information about them.
Thank you for this comment, we agree that our IHC discussion sections were lacking and we have now added the following sentences:
Line: 590-602: Incredibly, results obtained from our immunohistochemistry studies demonstrate the ability of Glyz to abolish increased expression of HMGB-1 within the endothelium, media, and adventitia in AA rings either exposed to Hcy or from rabbits fed a 4-week AD. Importantly, this finding suggests that Glyz may be able to enter cells and interfere with production and degradation steps that are central to HMGB-1 biology and release. This hypothesis is supported by a study involving cerebral vasospasm secondary to subarachnoid hemorrhage in rats, which reported a reduction of intracellular peptide and mRNA expression of HMGB-1 in rat basilar artery after Glyz treatment [47]. Similarly, a complementing study has shown that in murine macrophages (RAW 264.7) challenged with lipopolysaccharide (a TLR4 activating lipoglycan present on the outer membrane of Gram-negative bacteria), treatment with Glyz was able to prevent shuttling of HMGB-1 from the nucleus into the cytoplasm and blocked upregulation of HMGB-1 mRNA [46].
Line 623-632: Reduced eNOS expression may explain decreased Ach sensitivity and subsequent impaired vasodilation, as eNOS peptide expression was significantly lower in rings exposed to Hcy or rhHMGB-1. Interestingly, we show that treatment with Glyz was able to enhance eNOS expression in the endothelium and adventitia of AA rings after Hcy incubation but not in AA rings from AD rabbits. These results align with a study that noted the anti-atherosclerotic properties of Glyz in male New Zealand White rabbits fed a 4-week high cholesterol (1 %) and fat (6 %) diet before being reverted to a 4-week low cholesterol (0.025 %) diet to induce atherosclerosis, as rabbits with Glyz-eluting stents fitted on the left carotid artery demonstrated increased eNOS expression, promoted reendothelialization, enhanced luminal diameter and inhibited intimal hyperplasia [55].
Line 681-704: Results from our immunohistochemistry experiments demonstrate that inhibiting HMGB-1 with Glyz was able to drastically reduce the expression of nitrotyrosine in all layers of AA rings incubated with Hcy and in the endothelium and media of rabbits fed a 4-week AD to promote HHcy-induced atherogenesis. This may be attributed to Glyz reducing oxidative and cell stress and promoting antioxidative mechanisms that attenuate nitrotyrosine formation. In male CD1 mice that underwent spinal cord surgery to produce injury at the thoracic level via extradural compression, pre-treatment (30 min before spinal cord injury) and post-treatment (1 hr and 6 hr after spinal cord injury) with Glyz (10 mg/kg) resulted in significantly reduced nitrotyrosine expression in the spinal cord 24 hr after injury [58]. Similar results have also been reported in mice with hepatotoxicity induced by acetaminophen, as treatment with Glyz (80 mg/kg) 90 min after acetaminophen administration was able to provide hepatoxicity protection by decreasing hepatic mitochondrial and hepatocellular damages (indicated by serum analysis of glutamate dehydrogenase, alanine aminotransferase and aspartate aminotransferase activity), attenuated mitochondrial swelling and distortion, reducing necrosis of hepatic cells, inhibiting neuronal nitric oxide synthase activity and mRNA expression and decreased liver protein tyrosine nitration and nitrotyrosine levels [59]. Furthermore, Glyz treatment (10 mg/kg) 5 min prior to carrageenan pleural cavity injection to induce pleurisy was able to attenuate lipid peroxidation and immune cell infiltration and decreased upregulation of intracellular adhesion molecules (e.g., P-selectin and intracellular adhesion molecule 1) and nitrotyrsoine in lung tissue [60]. While the ability of Glyz to attenuate Hcy-induced pathologies has yet to be validated, results from our study are supported by others that have determined Glyz’s ability to alter nitrotyrosine expression making it a promising therapeutic option for CVDs.
Line 721-742: The anti-inflammatory effect of Glyz is well established and is shown by multiple studies highlighting using models in which its ability to suppress the expression immune receptors (e.g., TLR2/4 and RAGE) and their downstream signaling molecules [23]. Results obtained from immunohistochemistry studies show the ability of Glyz to potently reduce expression of TNFα in the different layers of AA rings incubated with Hcy or from rabbits fed a 4-week AD. Similar result shave been reproduced in diabetic rats with atherosclerosis (induced by high fat diet and low dose streptozotocin), as a 16-week treatment of Glyz was able to significantly reduce serum and aorta levels TNFα and other pro-inflammatory cytokines (i.e., interleukin-1β and -6) as well as decreased atherosclerotic plaque lesion size and intima thickness [66]. Conducive results have also been demonstrated in rats with coronary microembolization-induced injury, as pre-treatment with Glyz ameliorated myocardial dysfunction by preventing myocardial necrosis, downregulating mRNA expression of HMGB-1, inducible nitric oxide synthase, tumor necrosis factor alpha, interleukin -1β and -6 and suppressed stimulation of the HMGB-1/TLR4/nuclear factor kappa beta pathway [67]. While the relationship between HMGB-1, Glyz and Hcy remains poorly understood, there is evidence that Glyz could be beneficial during Hcy-induced pathologies [68]. A study utilizing HUVECs (CRL-1730) noted that Hcy directly causes endothelial cell injury by effecting lectin-like oxidized-low-density lipoprotein receptor-1 DNA methylation through stimulation of TLR4/nuclear factor-kappa beta/DNA methyltransferase [68]. While the authors did not investigate production of pro-inflammatory cytokines, this study provides a rationale that TLR4 may have a prominent role in Hcy-induced endothelial cell dysfunction by activating TLR4 and resulting in the release of TNFα.
Comment #5: As a minor detail, the name of the plant Glycyrrhiza glabra should appear in italics.
Thank you for bringing this mistake to our attention. The words Glycyrrhiza glabra have now been written in italics in the manuscript.
Round 2
Reviewer 2 Report
The authors have addressed all the comments requested. On behalf of this reviewer, this manuscript can be accepted in the present form.
It is worth investing time in a final proofreading for minor details.